# Interpretable Self-Supervised Learning via Representer Landmarks and Nyström Approximation

**Maedeh Zarvandi** [1 2]   **Michael Timothy** [1]   **Theresa Wasserer** [1]   **Debarghya Ghoshdastidar** [1 2]

## Abstract

Self-supervised learning (SSL) learns representations from massive unlabeled data, yet the resulting models typically operate as black boxes, necessitating domain-specific explanations. We introduce KREPES, a unified framework to analytically interpret the learned representations of SSL objectives, including SimCLR, BYOL, and VICReg. By bridging empirical neural tangent kernel approximations of neural networks with the Representer Theorem for kernels, we express the learned latent space directly via "Representer Landmarks", which are the representations of influential unlabeled training examples. We introduce novel metrics, "Sample-Specific Influence Score", "Concept-Conditioned Influence Score" and "Feature Alignment Gap", to quantify the transparency of the learned representations. KREPES enables direct audit of the latent space without supervision, for example, revealing an algorithmic bias in the Adult-1M dataset where SSL uses demographic proxies for income. Finally, to ensure scalability to benchmarks with 1M+ samples (ImageNet-1K, Adult-1M), KREPES introduces a novel Nyström approximation-based analytical inference framework for SSL objectives.

## 1. Introduction

Representation learning has shifted from handcrafted features like SURF (Park et al., 2022) to learned representations that generalize across downstream tasks, with Self-Supervised Learning (SSL) being the dominant paradigm, using massive unlabeled datasets to drive the discovery of meaningful semantic structures (Devlin et al., 2019; Caron et al., 2021; Arik & Pfister, 2021). However, the complexity required to capture these structures makes modern SSL models opaque. The community has tried to address this by adapting post-hoc explanations such as saliency maps (Simonyan et al., 2014; Selvaraju et al., 2017) and linear probes (Alain & Bengio, 2017). More recent works have proposed domain-specific interpretable SSL architectures, including geometric bottlenecks for video pose estimation (Jakab et al., 2020), decoding biological prototypes for single-cell transcriptomics (Golpayegani et al., 2025), and modular architectures still reliant on saliency-based analysis (Seong & Shim, 2024). However, these approaches remain tied to specific domains or downstream tasks and do not explain the internal mechanics of SSL representations in the sense of inherently interpretable models advocated by Rudin (2019). We address this gap by introducing a principled attribution framework for interpreting self-supervised representations.

Defining interpretability in SSL poses a fundamental challenge: self-supervised pre-training, a.k.a. the representation learning step, does not involve specific prediction tasks or ground-truth labels. Hence, typical feature attribution-based explainability is ill-defined for SSL. Sample-based interpretability offers a natural resolution to this paradox. By tracing the embedding of a new sample back to specific training instances, one can explain the learned representation through data influence rather than label prediction. The theory of reproducing kernels, specifically the Representer Theorem (Schölkopf et al., 2001), provides a mathematical framework to achieve interpretability by expressing the model output as a weighted sum of similarities to training samples, referred to as "representer points". Recent work has adapted this idea to derive interpretable approximations of supervised neural networks (NNs) via kernels (Yeh et al., 2018; Tsai et al., 2023; Engel et al., 2023). These interpretable models strongly rely on the supervised learning setting, for example, by using the analytical solution of kernel regression as the model output. Hence, the problem of interpretable representation learning in an unsupervised or self-supervised setting remains unaddressed.

Despite the potential of kernel approximations and Representer Theorem for deriving interpretable SSL models, this has not yet been explored in the current literature on kernel

---

[1]Technical University of Munich, TUM School of Computation, Information and Technology [2]Munich Center for Machine Learning (MCML). Correspondence to: Maedeh Zarvandi <maedeh.zarvandi@tum.de>.

*Proceedings of the $43^{rd}$ International Conference on Machine Learning*, Seoul, South Korea. PMLR 306, 2026. Copyright 2026 by the author(s).

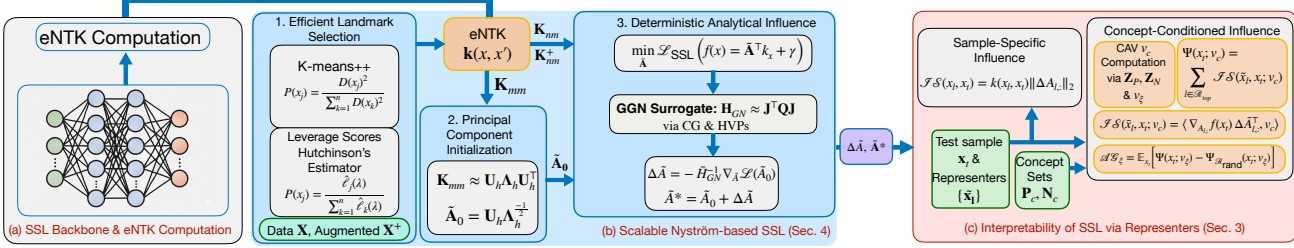

*Figure 1.* **Overview of the KREPES framework for Interpretable SSL. (a) SSL Backbone & eNTK Computation:** The pipeline begins by approximating a frozen, pretrained deep neural network via its eNTK. **(b) Scalable Nyström-based SSL (Sec. 4):** We isolate the influence of the SSL objective by computing a single deterministic Generalized Gauss-Newton step ($\Delta\tilde{A}$) away from the unsupervised Nyström manifold ($\tilde{A}_0$), solved efficiently via Conjugate Gradient and HVPs. **(c) Interpretability of SSL via Representers (Sec. 3):** The coefficient ($\Delta\tilde{A}$) unlocks rigorous latent space auditing. We derive Sample-Specific Influence Scores and Concept-Conditioned Influence (via CAV) to explain why a test sample is mapped to its specific representation.

theory for SSL models (Cabannes et al., 2023; Simonyan et al., 2014; Esser et al., 2024; Fleissner et al., 2025). This is not surprising because kernel methods are computationally ill-suited in the regime of SSL. While SSL models require large amount of unlabeled training data to learn generalizable features (Radford et al., 2021), standard implementations of kernel methods scale poorly with training sample size (quadratic in memory and cubic in runtime). The computational bottleneck of kernel machines has been addressed in the context of kernel regression through efficient solvers based on Nyström approximation and random Fourier features (Meanti et al., 2020; Abedsoltan et al., 2023), with some recent work extending the techniques to other convex loss functions (Della Vecchia et al., 2024). However, these efficient kernel methods cannot be directly used in the context of SSL, where the loss function is generally nonconvex, for instance, SimCLR, Barlow twins, BYOL, etc. (Chen et al., 2020; Grill et al., 2020a; Zbontar et al., 2021).

**Our Contribution.** The objective of this paper is to develop a unified framework for constructing interpretable representations, pretrained with SSL objectives. The proposed KREPES pipeline broadly uses the following approach: (i) a self-supervised neural network is approximated by its empirical neural tangent kernel (eNTK); (ii) we leverage Representer Theorem to express the representation learned by a (kernel) SSL model in terms of few unlabeled pretraining samples; (iii) we analytically infer the dual coefficients by projecting the non-convex SSL objective onto a finite-dimensional Nyström subspace and computing a single, deterministic Generalized Gauss-Newton step. KREPES includes several novel concepts and key technical innovations, such as:

**1) Representer Landmarks.** Influential samples that characterize embedding space learned by SSL models.

**2)** New metrics, **Sample-Specific Influence Score**, **Concept-Conditioned Influence Score** and **Feature Alignment Gap.** To quantify the transparency of learned representations. Our metrics help to extract feature importance

rankings for SSL models.

**3) Scalable Kernel SSL Attribution.** We leverage a Nyström approximation to scale deterministic analytical influence derivations to 1M+ samples for non-convex SSL objectives via a single Generalized Gauss-Newton step, incorporating landmark selection. Empirical results demonstrate its efficacy in feature importance extraction and fairness auditing, while validating eNTKs as reliable proxies for deep NNs. Our framework is illustrated in Figure 1.

## 2. Background on Interpretability and Representation with Kernels

**Representer Theorem for Interpretability.** The use of kernels for interpretability is grounded in the classical Representer Theorem for reproducing kernels (Kimeldorf & Wahba, 1970), which states that the minimizer $f^*$ of a regularized empirical risk on a Reproducing Kernel Hilbert Space (RKHS) $\mathcal{H}$ can be expressed as a linear combination of the training data kernels: $f^*(x) = \sum_{i=1}^{n} \alpha_i k(x_i, x)$. In the context of interpretable machine learning, this theorem allows *sample-based explanations*, where a model's prediction on a test sample $x_t$ is decomposed into the influential contributions of training samples (prototypes). Yeh et al. (2018) adapted this framework to explain deep neural networks (DNNs) by treating the pre-activation output of the final layer as a linear model operating on the feature space learned by the previous layers. Using the stationary condition of the empirical risk minimization (where the gradient of loss with respect to weights is zero at optimality), they decompose the DNN output $f(x_t)$ as $f(x_t) = \sum_{i=1}^{n} \alpha_i k(x_t, x_i), \ k(x_t, x_i) = \Phi(x_t)^\top \Phi(x_i)$, where $\Phi(x_t)$ denotes the feature representations of the sample $x_t$. The coefficients $\alpha_i$ are analytically derived from the loss gradients, specifically $\alpha_i \propto \dfrac{\partial L(x_i, y_i)}{\partial f(x_i)}$, serving as a measure of the influence of the training sample $x_i$. To bypass the analytical derivation, Engel et al.

(2023) introduced *Kernel General Linear Models* (kGLM) as learned surrogates. They define $\text{kGLM} : \mathcal{X} \rightarrow \mathbb{R}^C$ as $\text{kGLM}(x) := W\, k(x, X) + b$, where $C$ denotes the number of classes, $W \in \mathbb{R}^{C \times N}$ and $b \in \mathbb{R}^C$ are learnable parameters optimized via cross-entropy on the same dataset as the neural network. This effectively learns the "weights" of influence directly. They define the attribution of training point $x_i$ to $x_t$ for $c \in [C]$ as $A(x_t, x_i)_c := W_{c,i}\, k(x_t, x_i) + \dfrac{b_c}{N}$, ensuring that sum of attributions equals the total logit score: $\sum_{i=1}^{N} A(x_t, x_i)_c = \text{kGLM}(x_t)_c$.

Parallel to this, Tsai et al. (2023) identify a sparse set of influential "representer points" for regularized high-dimensional models. Introducing:(i) a *global importance* term derived from the loss gradient and regularization sub-gradients, (ii) a *local importance* term capturing similarity $k(x_i, x_t)$. This separation is a concept we adapt in our influence metrics.

**Representation Learning via SSL.** In contrast to the aforementioned works on interpreting supervised DNNs, we focus on the setting of representation learning via SSL. In this setting, one has access to $n$ unlabeled samples from an input space $\mathcal{X}$, potentially with different augmented views. We denote the samples as $\{(x_i^1, \ldots, x_i^p)\}_{i=1}^n$ representing $p$ augmented views per sample. While unsupervised autoencoder use only one view ($p = 1$), SSL principles of contrastive learning, joint embedding, self-distillation, etc. use multiple views ($p \geq 2$). The general goal of this approaches is to learn an embedding $f : \mathcal{X} \rightarrow \mathbb{R}^h$ by minimizing a loss function $\mathcal{L}$ defined using augmented data

$$\min_{f \in \mathcal{F}} \mathcal{L}\left(\left\{f(x_i^j)\right\}_{i \in [n], j \in [p]}\right), \qquad (1)$$

where $\mathcal{F}$ denotes the hypothesis class, typically parameterized by DNNs. A plethora of loss functions $\mathcal{L}$ have been proposed, such as SimCLR (Chen et al., 2020) and Barlow Twins (Zbontar et al., 2021). SSL loss functions implemented in KREPES are discussed in appendix D.

**Kernel Model for SSL Representations.** To address interpretability in SSL via Representer Theorem, we need to characterize $\mathcal{F}$ in terms of kernel models. The following model has been used to derive statistical guarantees for SSL (Cabannes et al., 2023; Esser et al., 2024), but its practical utility was not previously explored. Let $k : \mathcal{X} \times \mathcal{X} \rightarrow \mathbb{R}$ be a positive definite kernel with associated RKHS $\mathcal{H}$, and feature map $\phi_x \in \mathcal{H}$, satisfying the reproducing property $k(x, x') = \langle \phi_x, \phi_{x'} \rangle$. $\mathcal{L}$ in (1) is optimized over the class $\mathcal{F}$ of linear operators from $\mathcal{H}$ to representation space $\mathbb{R}^h$

$$\mathcal{F} = \left\{ f(x) = W\phi_x \mid W = [w_1, \ldots, w_h]^\top, w_i \in \mathcal{H} \right\}.$$

Here, the learnable map $W : \mathcal{H} \rightarrow \mathbb{R}^h$ is parameterized by rows $w_1, \ldots, w_h \in \mathcal{H}$, such that the $i$-th dimension of the learned embedding corresponds to $\langle w_i, \phi_x \rangle$. The Representer Theorem (Kimeldorf & Wahba, 1970; Schölkopf

et al., 2001) shows that it suffices to optimize $\mathcal{L}$ over $f(x) = \overline{W}\phi_x$, where each row of $\overline{W}$ lies in the span of $\left\{\phi_{x_i^j}\right\}_{i \in [n], j \in [p]}$. Hence, the optimization (1) can be done over a finite dimensional space $\mathcal{F} \subseteq \mathbb{R}^{h \times np}$

$$\mathcal{F} = \left\{ f(x) = \overline{W}\phi_x = \sum_{i \in [n], j \in [p]} \alpha_i^j k(x_i^j, x) \right\} \qquad (2)$$

where $\{\alpha_i^j\}_{i \in [n], j \in [p]} \in \mathbb{R}^h$. Additional Tikhonov regularization or orthogonality constraints ensure that the optimum over the above restricted space is the unique optimum over the RKHS $\mathcal{H}$ (Schölkopf et al., 2001; Esser et al., 2024).

## 3. Interpretability of SSL via Representers

Our objective is to obtain *interpretable representations*, that is, embeddings where the mapping of a test sample $x_t \in \mathcal{X}$ to a specific location in the latent space can be attributed to the influence of training samples. To achieve this, we propose an attribution framework that bridges deep SSL models with influence functions (Weisberg & Cook, 1982). Our pipeline consists of three primary steps. First, we capture the model's dynamics by computing its eNTK, with the scalable derivation detailed in Appendix A. Second, leveraging this kernel, we calculate the influence scores for a set of training landmarks, effectively quantifying how individual instances shape the overall self-supervised latent space by minimizing any of the SSL objectives discussed in section 2. Finally, we extend this to the inference stage by computing a per-sample influence score which traces the representation of $x_t$ back to its most influential landmarks. While prior approaches leveraging either the Representer Theorem or influence functions have been confined to supervised settings (Yeh et al., 2018; Tsai et al., 2023; Koh & Liang, 2017; Engel et al., 2023), we are the first to bridge these two paradigms to derive a rigorous attribution framework for self-supervised learning. We next detail the proposed method and present our interpretability scores.

### 3.1. Sample-Specific Influence Score

Classical influence function $IF(z; T, P)$ (Hampel, 1974), measures the infinitesimal change in a statistical functional $T(P)$ under perturbation of the data distribution $P$ by a point mass $\delta_z$, via Gâteaux derivative in direction of the mixture distribution $P_{\epsilon, z} := (1 - \epsilon)P + \epsilon\delta_z$

$$IF(z; T, P) = \lim_{\epsilon \to 0} \frac{T(P_{\epsilon, z}) - T(P)}{\epsilon}. \qquad (3)$$

Setting $T(\hat{P}_n) := \hat{\theta} = \arg\min_{\theta \in \Theta} \mathcal{L}(\hat{P}_n, \theta)$, where $\hat{P}_n = \dfrac{1}{n} \sum_{i=1}^{n} \delta_{x_i}$ is the empirical distribution, Koh & Liang (2017) approximated 3 via a first-order Taylor expansion of the

optimality condition $\nabla_\theta \mathcal{L}(\hat{P}_{n,\epsilon}, \hat{\theta}_\epsilon) = 0$, under the strict regularity conditions $\mathcal{L}(x, \cdot) \in \mathcal{C}^2(\Theta)$ and Hessian $H_{\hat{\theta}} = \nabla_\theta^2 \mathcal{L}(\hat{P}_n, \hat{\theta}) \succ 0$, yielding

$$IF(z; \hat{\theta}, \hat{P}_n) = -H_{\hat{\theta}}^{-1} \nabla_\theta \mathcal{L}(z, \hat{\theta}). \quad (4)$$

In Appendix E, we show that we can adapt this to self-supervised objectives s.a. Barlow Twins, even though their population loss is not of the form $\mathbb{E}_P[\ell(X, \theta)]$, rather a differentiable function of expectation-type statistics of $P$. Thus, under contamination, these expectations have Gâteaux derivatives, and the differentiability permits the chain rule.

Applying this to deep SSL models is intractable due to their nonlinearity and non-convex loss landscape. KREPES addresses this using eNTK and interpretability via Representer Theorem (Yeh et al., 2018; Tsai et al., 2023), which attributes predictions to representer points via dual coefficients. The linear representation $f(x_t) = \sum_{l=1}^n k(x_l, x_t) A_{l,:}$ (Section 2), assigns the influence of $x_l$ to the magnitude of its corresponding coefficient row $A_{l,:}$.

Classical $IF$ uses the point-contamination direction $\frac{d}{d\epsilon} \nabla_\theta \mathcal{L}(\theta(P), P_{\epsilon,z})\big|_{\epsilon=0}$. Here, our goal is different: we measure the local displacement induced by the SSL objective relative to the kernel-geometric initialization $A_0$.

To obtain a finite-dimensional analogue of the influence step, we construct a locally convex surrogate in the representer parameter $A$. The Hessian in (4) is the regularized GGN curvature $\bar{H}_{GN} = H_{GN} + \lambda I \succ 0$, evaluated at $A_0 = U_h \Lambda_h^{-1/2}$. The resulting second-order surrogate w.r.t the global parameter step $\Delta A \in \mathbb{R}^{n \times h}$ is $\tilde{\mathcal{L}}(\Delta A) = \mathcal{L}(A_0) + \text{Tr}(\nabla_A \mathcal{L}(A_0)^\top \Delta A) + \frac{1}{2} \text{vec}(\Delta A)^\top \bar{H}_{GN} \text{vec}(\Delta A)$. Solving $\nabla_{\Delta A} \tilde{\mathcal{L}} = 0$ yields

$$\text{vec}(\Delta A) = -\bar{H}_{GN}^{-1} \text{vec}(\nabla_A \mathcal{L}(A_0)). \quad (5)$$

Unlike supervised influence functions (Koh & Liang, 2017), SSL requires quantifying the induced shift in the test representation $f(x_t)$, which due to our linear formulation using *Representer Theorem*, admits the exact landmark-wise decomposition $\Delta f(x_t) = f(x_t; A_0 + \Delta A) - f(x_t; A_0) = \sum_{l=1}^n k(x_l, x_t) \Delta A_{l,:}^\top$, where the contribution of landmark $x_l$ is $\nabla_{A_{l,:}} f(x_t) = k(x_l, x_t) I_h$. We define *Sample-Specific Influence Score*

$$\mathcal{IS}(x_l, x_t) = \text{sgn}(k(x_l, x_t)) \left\| \nabla_{A_{l,:}} f(x_t) \Delta A_{l,:}^\top \right\|_2, \quad (6)$$

which is equivalent to $\mathcal{IS}(x_l, x_t) = k(x_l, x_t) \|\Delta A_{l,:}\|_2$. Let $\omega_l = \left\| \Delta A_{l,:} \right\|_2$ denote contribution of the $l$-th representer to the projection space. To assess the semantic coverage of these representers, let $\pi$ be a permutation of indices such that $\omega_{\pi(1)} \geq \omega_{\pi(2)} \geq \cdots \geq \omega_{\pi(n)}$. We define *Class Coverage* $\kappa_\mathcal{C}$ as the minimum number of top-ranked representers

required to span the set of all ground-truth classes $Y$

$$\kappa_\mathcal{C} = \min \left\{ \kappa \in [n] \,\middle|\, Y \subseteq \bigcup_{i=1}^\kappa \{y(x_{\pi(i)})\} \right\}. \quad (7)$$

### 3.2. Concept-Conditioned Influence Score

We extend our representer framework to interpret representations through a concept-alignment lens. We quantify the degree to which semantic concepts, embodied by influential representers, drive the prediction for a test sample $x_t$.

**Concept Activation Vector (CAV).** Let $c$ be a concept of interest (e.g., "Sea"). We construct a positive concept set $P_c = \{x \mid x \in \text{concept } c\}$ and a negative set $N_c$ consisting of randomly sampled non-concept examples. We compute their representations in the learned kernel space as $Z_P = A^\top K_{P,n}^\top$, $Z_N = A^\top K_{N,n}^\top$, where $K_{P,n} \in \mathbb{R}^{|P_c| \times n}$ and $K_{N,n} \in \mathbb{R}^{|N_c| \times n}$ denote the kernel matrices between the concept sets and the learned representers. Following Kim et al. (2018), we derive the *CAV* $v_c \in \mathbb{R}^h$ as the normalized orthogonal vector to the separating hyperplane, pointing in the direction of the concept.

We define *Concept-Conditioned Influence Score* of representer $x_l$ on a test sample $x_t$ w.r.t. concept $c$ as the projection of the representer's contribution onto the concept vector

$$\mathcal{IS}(x_l, x_t; v_c) = \langle \nabla_{A_{l,:}} f(x_t) \Delta A_{l,:}^\top, v_c \rangle \quad (8)$$

This attributes the prediction to representers that are both influential for $x_t$ and aligned with concept $c$. Consequently, the aggregated concept influence is defined as the sum of the scores over top influential representers ($\mathcal{R}_{\text{top}}$)

$$\Psi(x_t; v_c) = \sum_{l \in \mathcal{R}_{\text{top}}} \mathcal{IS}(x_l, x_t; v_c) \quad (9)$$

A positive $\Psi(x_t; v_c)$ indicates that the selected representers steer the representation of $x_t$ towards concept $c$, whereas a negative value indicates concept suppression. Details and algorithm are provided in Appendix J.

**Feature Importance via Alignment Gap.** We audit tabular domains through feature alignment, where an input feature $\xi$ itself is a semantic concept. For feature $\xi$, we define feature agreement between $x_t$ and $x_l$, as $v_\xi(x_t, x_l) = 1 - \min\left(\frac{|x_{t,\xi} - x_{l,\xi}|}{\Delta_\xi}, 1\right)$, where $\Delta_\xi = \max(\mathcal{X}_\xi) - \min(\mathcal{X}_\xi)$ denotes the dynamic range of $\xi$ across the dataset. *Feature-Conditioned Influence Score* is then

$$\mathcal{IS}(x_l, x_t; v_\xi) = \left\| \nabla_{A_{l,:}} f(x_t) \Delta A_{l,:}^\top \right\|_2 v_\xi(x_t, x_l), \quad (10)$$

and the aggregated feature influence is $\Psi(x_t; v_\xi) = \sum_{l \in \mathcal{R}_{\text{top}}} \mathcal{IS}(x_l, x_t; v_\xi)$. Larger values indicate stronger preservation of $\xi$, whereas values close to zero indicate weak

alignment. To quantify global feature importance, we define *Feature Alignment Gap*

$$\mathcal{AG}_\xi = \mathbb{E}_{x_t}[\Psi(x_t; v_\xi) - \Psi_{\mathcal{R}_{\text{rand}}}(x_t; v_\xi)], \qquad (11)$$

where $\Psi_{\mathcal{R}_{\text{rand}}}$ denotes aggregation over random samples. Positive $\mathcal{AG}_\xi$ indicates that the SSL latent geometry preferentially preserves feature $\xi$, whereas negative $AG_\xi$ indicates relative suppression of feature $\xi$ in the landmark set. This reveals implicit biases in the SSL latent geometry, e.g., dominance of demographic proxies, without label supervision.

# 4. Scalable Nyström-based SSL

Naively evaluating the Newton step (5) with a SSL objective, on the full kernel, due to quadratic memory and cubic time costs, is intractable. For kernel ridge regression (KRR), Rudi et al. (2017) showed Nyström approximation attains the same statistical performance up to logarithmic factors, with $O(n\sqrt{n})$ time and $O(n)$ memory complexity. We use this as motivation for the Nyström parameterization in SSL.

The Nyström approximation used in Rudi et al. (2017); Abedsoltan et al. (2023) further restricts the search space $\mathcal{F}$ based on the notion of a General Kernel Model (GKM) that corresponds to function $f : \mathcal{X} \to \mathbb{R}^h$ of the form $f(x) = \sum_{l=1}^m \beta_l k(x_l, x)$, where $\{x_l \in \mathcal{X}\}_{l=1}^m$ are landmarks, and $\{\beta_l \in \mathbb{R}^h\}_{l=1}^m$ are learnable parameters. Landmarks correspond to the entire dataset (all $np$ samples $x_i^j$ in our case) with $m \ll n$. Rudi et al. (2017) show that $m = O(\sqrt{n})$ suffices for optimal accuracy.

**Nyström Approximation in Representation Learning.** In the present context, where the empirical distribution consists of augmented samples, it is natural to select $m \ll n$ tuples $(x_i^1, \ldots, x_i^p)$ as landmarks. We denote this subset by $(\tilde{x}_i^1, \ldots, \tilde{x}_i^p)$ for $i \in [m]$. Consequently, we restrict our representations to the following class of functions:

$$\mathcal{F} = \left\{ f(x) = \sum_{i \in [m], j \in [p]} \tilde{\alpha}_i^j k(\tilde{x}_i^j, x) + \gamma \right\}$$

where $\{\tilde{\alpha}_i^j\}_{i \in [m], j \in [p]} \in \mathbb{R}^h$ are the dual coefficients, hence $\tilde{A} \in \mathbb{R}^{mp \times h}$, and $\gamma \in \mathbb{R}^h$ is a global shift vector. This parameterization projects the infinite-dimensional RKHS onto a finite-dimensional subspace $\mathbb{R}^{mp \times h}$.

This function class reformulates the non-convex SSL objective into a finite parameter space over $\{\tilde{\alpha}_i^j\}$, enabling analytical computation of the Newton step derived in Section 3.1. Furthermore, $\gamma$ captures the spatial centering inherent to contrastive SSL objectives—a geometric shift that cannot be expressed purely via kernel combinations of the landmarks. This parameterization enables KREPES to extend the influence framework to arbitrary self-supervised losses. Computational details are provided in Appendix B.

**Notation.** Let $k_x = [k(\tilde{x}_i^j, x)]_{i,j} \in \mathbb{R}^{mp}$ be the vector of kernel evaluations against all landmarks. We parameterize the learned function as $f(x) = \tilde{A}^\top k_x + \gamma$, with coefficients $\tilde{A} \in \mathbb{R}^{mp \times h}$ and bias $\gamma \in \mathbb{R}^h$. We define the full kernel matrices $K_{mm} \in \mathbb{R}^{mp \times mp}$ (landmarks vs. landmarks) and $K_{nm} \in \mathbb{R}^{np \times mp}$ (data vs. landmarks); the superscript $j$ denotes restriction to the $j$-th augmented view (e.g., $K_{mm}^j \in \mathbb{R}^{m \times m}$). Regularization constraints on the RKHS norm $\|W\|_{\mathcal{H}}^2$ and orthogonality $WW^\top = I$ translate to $\text{Tr}(\tilde{A}^\top K_{mm}\tilde{A})$ and $\tilde{A}^\top K_{mm}\tilde{A} = I$, respectively. $\odot$ denotes the Hadamard product.

## 4.1. Principal Component Initialization

We propose a principled parameter initialization scheme for $\tilde{A}$. We begin with the standard Nyström approximation $K_{nn} \approx K_{nm}K_{mm}^\dagger K_{mn}$, (Williams & Seeger, 2000) where $K_{mm}^\dagger$ denotes the Moore–Penrose pseudoinverse of $K_{mm}$. Let the rank $h$ truncated eigendecomposition of $K_{mm}$ be $K_{mm} \approx U_h \Lambda_h U_h^\top$, with $U_h \in \mathbb{R}^{m \times h}$ the top h eigenvectors and $\Lambda_h \in \mathbb{R}^{h \times h}$ the corresponding eigenvalues, the pseudoinverse is then $K_{mm}^\dagger \approx U_h \Lambda_h^{-1} U_h^\top$. Substituting this into the Nyström approximation and defining $\Phi = K_{nm}U_h\Lambda_h^{-1/2} \in \mathbb{R}^{n \times h}$, yields the following form for the low-rank kernel approximation

$$K_{nn} \approx (K_{nm}U_h\Lambda_h^{-1/2})(K_{nm}U_h\Lambda_h^{-1/2})^\top = \Phi\Phi^\top \quad (12)$$

Consider our parameter $\tilde{A} \in \mathbb{R}^{m \times h}$ as a projection from kernel space into a $h$-dimensional feature space, such that $f(\mathcal{X}) = K_{nm}\tilde{A} \in \mathbb{R}^{n \times h}$; comparing with the Nyström feature map $\Phi$, we observe that a natural initialization is $\tilde{A}_0 = U_h\Lambda_h^{-1/2}$. By setting the Taylor expansion point (Section 3.1) to this $\tilde{A}_0$, we ground the convex surrogate on the unsupervised data manifold. Consequently, the Newton step $\Delta\tilde{A}$ isolates the influence induced by the SSL objective, decoupling it from the kernel covariance. Empirical validation of this initialization is provided in Appendix G.

## 4.2. Deterministic Analytical Influence via GGN

To evaluate the analytical influence derived in Section 3.1, we must compute the Newton step $\Delta\tilde{A} = -\bar{H}_{GN}^{-1}\nabla_{\tilde{A}}\mathcal{L}(\tilde{A}_0)$. Direct inversion of $\bar{H}_{GN} = H_{GN} + \lambda I$ incurs an intractable $\mathcal{O}(m^3)$ complexity. We bypass this by recasting the operation as the linear system $\bar{H}_{GN}\Delta\tilde{A} = -\nabla_{\tilde{A}}\mathcal{L}(\tilde{A}_0)$, which we solve efficiently via the Conjugate Gradient (CG). Since CG relies solely on matrix-vector multiplications, we avoid explicitly forming $\bar{H}_{GN}$ which requires $\mathcal{O}(m^3)$ memory. Instead, we compute exact global gradients and Hessian-Vector Products (HVPs) dynamically via forward-over-reverse automatic differentiation. To scale to 1M+ samples, each gradient/HVP is streamed over minibatches, so memory is independent of $n$ up to minibatch storage and Nyström parameters. The cost is linear in $n$ per

CG iteration, with dependence on $m$, $h$, and the number of CG iterations. Thus, KREPES approximates the analytically defined GGN step without forming $\bar{H}_{GN}$.

**Generalized Gauss–Newton (GGN) Approximation.** For non-convex objectives $\mathcal{L}(\theta)$, we approximate the Hessian $\nabla^2 \mathcal{L}$ with the PSD surrogate $H_{\text{GN}} = J^\top Q J$, where $J$ is the Jacobian of the model output and $Q$ encapsulates local convexity of the loss (Korbit et al., 2024). The exact HVP for an arbitrary vector $d$ is derived for specific objectives as follows (full derivations in Appendix F):

**1) Barlow Twins (Non-Linear Least Squares).** We recast the cross-correlation objective as the squared norm of a residual vector, transforming the optimization into a non-linear least squares problem:

$$\min_\theta \|r(\theta) = \text{vec}(W \odot (C - I))\|_2^2, \quad (13)$$

where $W_{ii} = 1$ and $W_{ij} = \sqrt{\lambda_{reg}}$ for $i \neq j$. The curvature simplifies to $H_{\text{BT}} = 2 J_r^\top J_r$. The HVP is evaluated analytically as:

$$\text{HVP}_{\text{BT}}(d) = 2 \cdot \text{vjp}(r, \theta, \text{jvp}(r, \theta, d)). \quad (14)$$

**2) SimCLR (Softmax Cross-Entropy).** We formulate Sim-CLR as a cross-entropy objective on row-wise logits $f(\theta)$. The curvature is defined as $H_{\text{SC}} = J_f^\top Q J_f$, where $Q$ is block-diagonal with per-row blocks $Q_i = \text{diag}(p_i) - p_i p_i^\top$ and $p_i = \sigma(Z_{i,:}) \in \mathbb{R}^C$. We compute the HVP without materializing the dense matrix $Q$ by exploiting the structured sparsity of the softmax Jacobian. Letting $u = \text{jvp}(f, \theta, d)$:

$$\text{HVP}_{\text{SC}}(d) = \text{vjp}(f, \theta, \, \mathbf{p} \odot u - \mathbf{p}(\mathbf{p}^\top u)). \quad (15)$$

### 4.3. Efficient Landmark Selection

Since Nyström is a low-rank approximation, the landmarks $\mathcal{Z} = \{\tilde{x}_l\}_{l=1}^m$ define the column space of $K_{nm}$ and the subspace for $\tilde{A}_0$ (4.1); hence, selecting them carefully ensures the initial projection captures the most informative directions in kernel space. We employ two selection strategies:

**K-means++ Seeding.** We sample landmarks iteratively proportional to their squared distance from the current set $\mathcal{Z} : P(x_j) \propto \min_{c \in \mathcal{Z}} \|x_j - c\|_2^2$ (Arthur & Vassilvitskii, 2007).

**Approximate Leverage Score Sampling.** Let $K \in \mathbb{R}^{n \times n}$, where $K_{ij} = \kappa(x_i, x_j)$. Leverage score for the $j$-th point is defined as $l_j(\lambda) = (K(K + \lambda n I)^{-1})_{jj}$ (Rudi et al., 2017); to circumvent the $O(n^3)$ complexity of exact inversion, we employ a randomized estimator. Let $\Pi \in \mathbb{R}^{n \times s}$ be a random sketching matrix with i.i.d. entries (Rademacher or $\mathcal{N}(0,1)$). The vector of leverage scores is approximated by

$$\hat{\ell}(\lambda) = \sum_{k=1}^s \Pi_{:,k} \odot (K z_k), \ \hat{\ell} \in \mathbb{R}^n, \text{ where } z_k \in \mathbb{R}^n \text{ is the}$$

solution to the linear system $K + \lambda n I) z_k = \Pi_{:,k}$ solved via

Conjugate Gradients. Landmarks are subsequently sampled with probability $P(x_j) = \frac{\hat{\ell}_j(\lambda)}{\|\hat{\ell}\|_1}$. More details and related ablations can be found in Appendix H.

## 5. Results and Discussion

In this section, we empirically validate KREPES's interpretability capacity on real-world benchmarks across both visual and tabular data, including ImageNet(1.2M), CIFAR-10(60k), FairFace(108k), MNIST(70k), Adult(1M), Covertype(1M), Higgs(1M), and Bank Marketing(45k). KREPES contains a diverse set of SSL objectives, demonstrating flexibility to integrate arbitrary losses. We first present our interpretability results via analyzing influence scores and feature auditing. Next, we present KREPES as a label-free model selection tool, then the faithfulness of eNTKs as surrogates for NNs.

**Sample-Specific Influence and Semantic Coverage.** Table 1 reports $\kappa_\mathcal{C}$ alongside downstream classification test accuracy. We observe that smaller $\kappa_\mathcal{C}$ values align with higher accuracy, indicating that the representations allocate high-norm samples across diverse categories. Remarkably, despite the absence of label supervision, the SSL models prioritize landmarks that span distinct semantic classes, reflecting an implicit alignment with downstream performance.

Figure 2 confirms semantic coherence of the learned influence on ImageNet. For diverse test samples (selected randomly for visualization from test set)—ranging from 'tench' to 'drilling platform'—the top-3 retrieved representer landmarks consistently belong to the same class. Therefore, despite the absence of label supervision during training, KREPES learns a latent space where the score in (6) aligns with human-understandable semantic similarity, rather than merely low-level texture or color statistics. Similar results are presented for CIFAR-10 in Appendix I.

**Concept-Conditioned Influence.** Figure 3 illustrates that the top-5 influential landmarks often exhibit strong alignment with the concept relevant to $x_{test}$, highlighting which landmarks drive the prediction. E.g., for concept Sea (from CIFAR-100), each landmark's alignment score shown at the bottom center, and the rightmost column reports the $\Psi_N(t, c)$ from (9) with $N = 5$. A positive score indicates that concept Sea supports the prediction, while a negative score reflects opposing influence. In the last row, the concept Sea is less evident in the test image, yielding a negative score as the model is trained on clearer examples of concept Sea. Landmarks from classes such as automobile contribute negatively (rows 3, 4, and 5), whereas airplane landmarks can contribute positively when there is sea in the image (row 1), or very blue sky (row 3). Details in the Appendix J.

**Quantitative Validation of Landmark Semantics vs. Sim-**

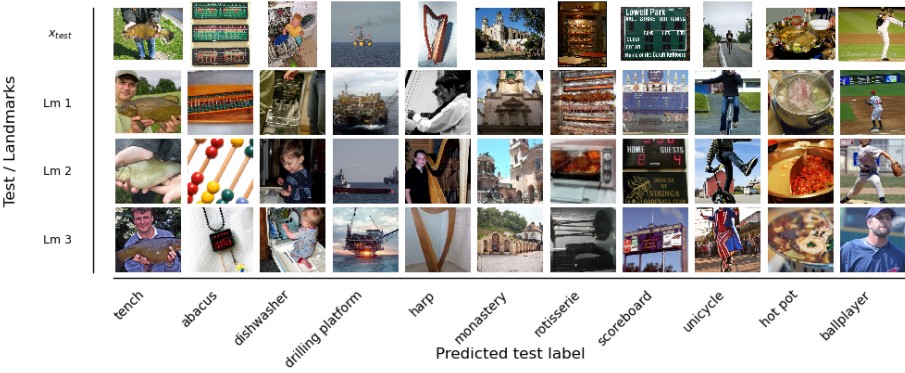

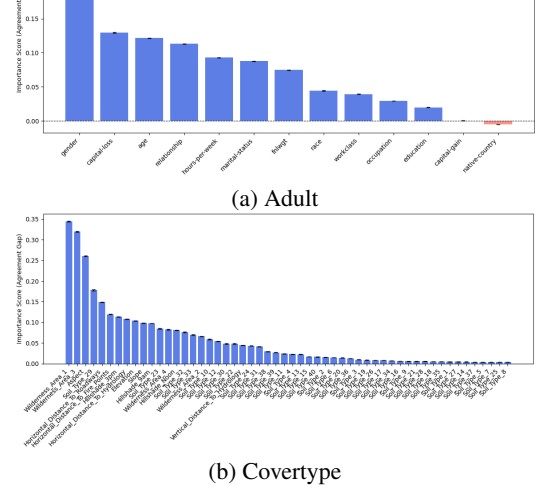

*Table 1.* Minimum number of landmarks required to cover all 10 CIFAR-10 classes ($\kappa_\mathcal{C}$) vs. Test Accuracy. Lower $\kappa_\mathcal{C}$ implies better semantic coverage.

| Loss Function | Acc (%) | $\kappa_\mathcal{C}$ |
|---|---|---|
| Barlow Twins | 91.18 | 12 |
| VICReg | 90.97 | 18 |
| BYOL | 90.56 | 26 |
| SimCLR | 90.36 | 27 |
| Spectral Cont. | 89.75 | 81 |

*Figure 2.* Sample-specific influential landmarks. First row displays test samples, followed by their top-3 influential landmarks, ranked by the influence score on ImageNet.

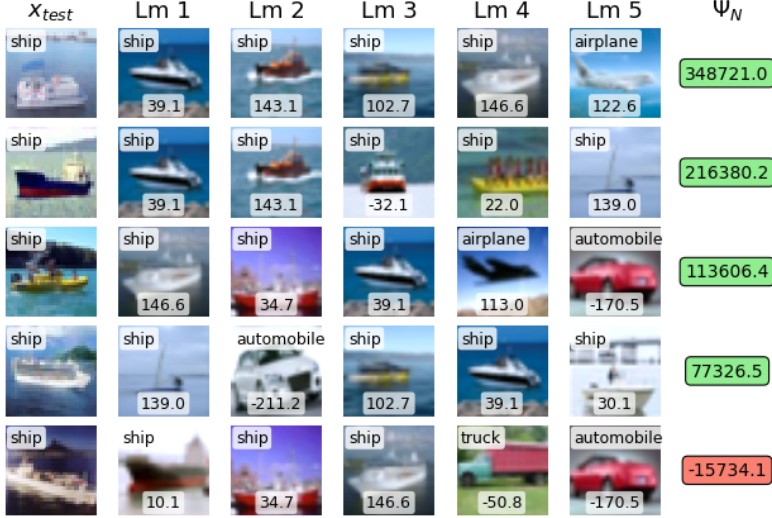

*Figure 3.* Influential landmarks: concept *Sea*. **Left:** $x_{test}$ with predicted label. **Middle:** top-5 influential landmarks with alignment scores (bottom). **Right:** aggregated score. Positive scores: concept *Sea* supports the prediction, negative scores oppose it. Less evident concept in 5th row, yields negative; automobile landmarks contribute negatively (rows 3, 4, 5), whereas airplane landmarks contribute positively when sea (row 1) or blue sky (row 3) is present.

*Figure 4.* **Feature Importance.** Alignment Gap computed over full test sets (Adult: $N = 200{,}000$; Covertype: $N = 116{,}202$, $K = 5$) with 95% confidence intervals. Tight error bars indicate negligible sampling variance; positive scores reflect high semantic relevance of features that dominate the SSL latent space.

*Table 2.* **Label Consistency Analysis.** KREPES vs. similarity-based baseline (nearest neighbors in SSL latent space) evaluated on KREPES's learned landmarks and full test sets. **Precision@K** measures the average percentage of top-$K$ influential landmarks sharing the test sample's class. **Majority@K** measures how often the majority class among the top-$K$ landmarks matches the prediction. **Hit Rate@K** measures the probability that at least one of the top-$K$ landmarks matches the correct class.

| | Precision@K | | | | Majority@K | | | | Hit Rate@K | | | |
|---|---|---|---|---|---|---|---|---|---|---|---|---|
| | Adult | | Cover | | Adult | | Cover | | Adult | | Cover | |
| $K$ | Ours | Base | Ours | Base | Ours | Base | Ours | Base | Ours | Base | Ours | Base |
| 1 | **0.872** | 0.809 | **0.772** | 0.550 | **0.872** | 0.809 | **0.772** | 0.550 | **0.872** | 0.809 | **0.772** | 0.550 |
| 2 | – | – | – | – | – | – | – | – | **0.948** | 0.809 | **0.914** | 0.734 |
| 3 | – | – | – | – | – | – | – | – | **0.970** | 0.968 | **0.944** | 0.826 |
| 5 | **0.854** | 0.798 | **0.702** | 0.495 | **0.901** | 0.842 | **0.777** | 0.509 | **0.987** | 0.968 | **0.983** | 0.903 |
| 10 | **0.844** | 0.789 | **0.659** | 0.471 | **0.921** | 0.872 | **0.827** | 0.567 | **0.997** | 0.990 | **0.990** | 0.948 |
| 20 | **0.832** | 0.778 | **0.617** | 0.449 | **0.910** | 0.868 | **0.777** | 0.501 | – | – | – | – |
| 50 | **0.813** | 0.762 | **0.555** | 0.429 | **0.885** | 0.862 | **0.687** | 0.413 | – | – | – | – |

**ilarity Baselines.** While standard SSL embeddings exhibit inherent semantic clustering, simple geometric proximity (e.g., nearest neighbors) lacks explanatory power and remains entangled with spurious structural artifacts. Table

2 quantitatively demonstrates that KREPES systematically outperforms a latent cosine-similarity baseline across all label consistency metrics. For instance, on the Adult dataset, KREPES achieves 87.2% Precision@1 compared to the

*Figure 5.* **Surrogate Faithfulness. Left (Accuracy Gap & Kendall-$\tau$):** The Accuracy Gap ($\Delta =$ KREPES $-$ NN) confirms KREPES preserves the representation quality of DNNs. Kendall-$\tau$ measures rank correlation between the predicted class probabilities of the NN and KREPES proving strict alignment of their decision boundaries. **Right (Confidence Drop):** Inference-Time Latent Ablation on frozen NN embeddings. Deleting the top 10 influential landmarks identified by KREPES collapses the prediction confidence of a $k$-NN ($k = 50$) classifier by orders of magnitude more than a random deletion, validating that KREPES isolates the true structural pillars of SSL manifold.

| Dataset (#samples) | Accuracy Gap ($\Delta$) / Kendall-$\tau$ | | | Confidence Drop (random / KREPES) | | |
| --- | --- | --- | --- | --- | --- | --- |
| | BT | SimCLR | VICReg | BT | SimCLR | VICReg |
| Adult (1M) | +0.06 / 0.845 | +0.12 / 0.842 | +0.12 / 0.840 | .0002 / **0.0572** | .0003 / **0.0567** | .0003 / **0.0483** |
| Higgs (1M) | +0.03 / 0.781 | -0.10 / 0.778 | +0.25 / 0.783 | .0003 / **0.0461** | .0004 / **0.0551** | .0003 / **0.0578** |
| ImageNet (1.2M) | -0.24 / 0.801 | -0.39 / 0.797 | -0.31 / 0.790 | .0001 / **0.0583** | .0002 / **0.0453** | .0002 / **0.0467** |
| CoverType (1M) | -0.41 / 0.872 | +0.87 / 0.861 | +0.47 / 0.863 | .0003 / **0.0810** | .0002 / **0.0731** | .0002 / **0.0771** |
| CIFAR-10 (60k) | -0.92 / 0.878 | -0.38 / 0.881 | -1.10 / 0.880 | .0011 / **0.0667** | .0010 / **0.0780** | .0012 / **0.0798** |

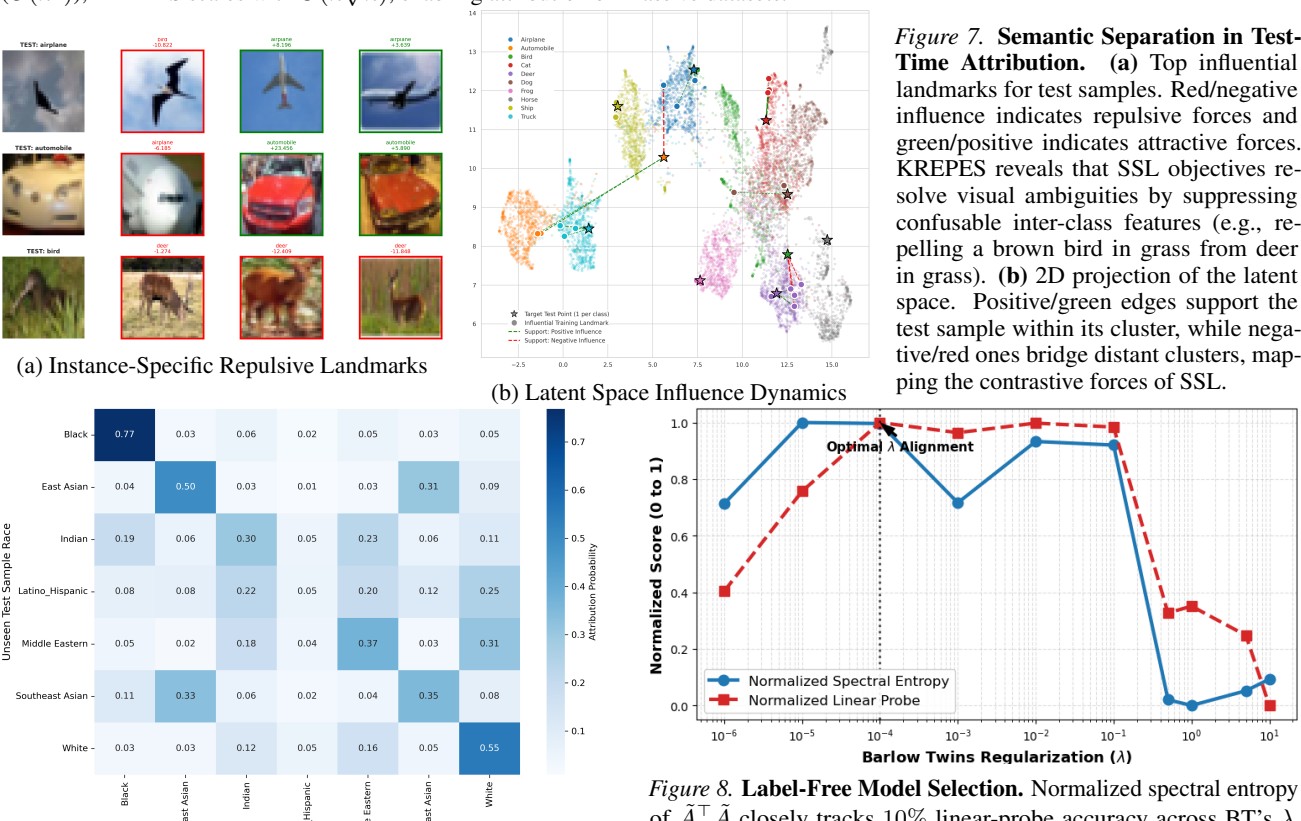

*Figure 6.* **Scalability and Performance Analysis.** We compare Nyström kernel against the Full Kernel on the Adult and Bank datasets. **Left (Accuracy):** KREPES stays in parity with the exact full kernel computation, showing negligible degradation in representation quality. **Right (Time):** Computation time (log-log scale) highlights the bottleneck. While the Full Kernel approach exhibits quadratic growth ($O(n^2)$), KREPES scales with $O(n\sqrt{n})$, enabling attribution on massive datasets.

(a) Instance-Specific Repulsive Landmarks

(b) Latent Space Influence Dynamics

*Figure 7.* **Semantic Separation in Test-Time Attribution.** **(a)** Top influential landmarks for test samples. Red/negative influence indicates repulsive forces and green/positive indicates attractive forces. KREPES reveals that SSL objectives resolve visual ambiguities by suppressing confusable inter-class features (e.g., repelling a brown bird in grass from deer in grass). **(b)** 2D projection of the latent space. Positive/green edges support the test sample within its cluster, while negative/red ones bridge distant clusters, mapping the contrastive forces of SSL.

*Figure 8.* **Label-Free Model Selection.** Normalized spectral entropy of $\tilde{A}^\top \tilde{A}$ closely tracks 10% linear-probe accuracy across BT's $\lambda$, with the entropy peak identifying the optimal $\lambda$ without labels.

*Figure 9.* **Cross-Demographic Representation Bias on FairFace.** Rows represent test samples, and columns their top-1 landmark. While in-group anchoring exists, KREPES exposes latent biases: Southeast Asians are anchored by East Asian landmarks, and Indians are conflated with Middle Eastern landmarks, revealing SSL objectives cluster disparate demographics based on spurious visual proxies.

baseline's 80.9%. This performance gap widens on the complex CoverType dataset (77.2% vs. 55.0% Precision@1). These results provides evidence that rather than merely retrieving geometrically proximal points, KREPES successfully isolates the dominant attribution drivers of the learned representation, establishing representer landmarks as rigorous, instance-specific auditing tools.

**Feature Importance & Fairness Auditing.** Figure 4 demonstrates KREPES's capacity for unsupervised auditing using the Alignment Gap. In Adult dataset (Figure 4a), we uncover a critical latent bias: sensitive demographic attributes like gender and relationship dominate the feature importance landscape, effectively dwarfing meritocratic features like education or occupation. This suggests the SSL model implicitly relies on demographic proxies despite the lack of supervision. In contrast, the Covertype analysis (Figure 4b) aligns with domain physics, where Wilderness Area and Elevation, key ecological determinants drive the representation, while rare soil types are correctly suppressed.

Beyond tabular domains, we apply KREPES to audit representation bias on FairFace. Since SSL relies on pixel-space augmentations without supervision, demographic conflation can emerge in the latent space. Using Sample-Specific Influence Scores, KREPES builds an unsupervised cross-demographic attribution matrix (Figure 9). While strong diagonals (e.g., 0.77 for Black samples) indicate in-group anchoring, prominent off-diagonal effects reveal representational bias induced by SSL objective. E.g., Southeast Asians are anchored to East Asian landmarks (33%), Indians are conflated with Middle Eastern (23%) and Latino/Hispanic (22%) landmarks relative to their own demographic (30%). Similarly, Middle Eastern samples are frequently anchored by White landmarks (31%). This shows KREPES audits biased geometries and spurious visual proxies.

**Semantic Separation in Test-Time Attribution.** While recent label-free influence methods focus on global self-influence over the training set (Harilal et al., 2024), KREPES enables test-time attribution by identifying the landmarks responsible for unseen embeddings. By modeling both positive and negative influence, KREPES reveals the repulsive dynamics of SSL objectives, where representations emerge not only from attraction between semantically similar samples, but also from suppression of visually confounding covariates. Figure 7a illustrates this effect. Negative scores (red) identify repulsive landmarks that separate confusable clusters: an airplane (a dark silhouette against sky) is repelled by a visually similar bird, while a white, rounded automobile is repelled by a visually analogous airplane fuselage. SSL objective suppresses these shared priors to embed the samples in the correct cluster, supported by positive influence landmarks (green). Most notably, a brown bird in tall grass exhibits strong negative influence

from deer landmarks sharing the same background statistics, demonstrating suppression of spurious texture and scene correlations. Figure 7b confirms this globally by projecting embeddings, landmarks, and influence vectors. Negative influence edges frequently connect distant latent clusters, revealing the boundary-separating dynamics of SSL representations beyond nearest-neighbor clustering.

**Computational Efficiency and Performance Validation.** A significant contribution of KREPES is making kernel methods feasible for SSL. Figure 6 validates the scalability of KREPES. While preserving the accuracy of the full kernel solution, KREPES reduces complexity from $O(n^2)$ to $O(n\sqrt{n})$, enabling scalable SSL interpretability.

**Label-Free Hyperparameter Tuning via Spectral Diagnostics.** Beyond attribution, KREPES intrinsically encodes structural properties of the representation space, offering a solution to a major limitation in SSL: zero-label model selection. Since $\tilde{A}$ captures the dual geometry of the data manifold, spectral diagnostics on $\tilde{A}^\top \tilde{A}$ can quantify dimensional collapse without supervision. Figure 8 tracks the normalized spectral entropy of $\tilde{A}^\top \tilde{A}$ against linear probe accuracy across a logarithmic grid of BT regularization $\lambda$ on MNIST. Notably, the entropy maximum aligns with peak accuracy, indicating that KREPES provides an unsupervised hyperparameter selection tool.

**eNTKs: Faithful Surrogates for NNs.** Since KREPES relies on eNTKs, we first verify that the kernel surrogate faithfully approximates NN dynamics. Table 5 validates this across three metrics. The accuracy gap ($\Delta = \text{Test Accuracy}_{\text{KREPES}} - \text{Test Accuracy}_{\text{NN}}$) shows that KREPES preserves downstream representation quality (details are in Appendix C). $Kendall - \tau$ correlation between NN and surrogate class probabilities confirms strong agreement of decision boundaries ($\tau \geq 0.84$ on Adult and CIFAR-10). Finally, we validate the attribution of the extracted landmarks via Inference-Time Latent Ablation. Using a $k$-NN classifier ($k = 50$) on NN embeddings. It shows that removing top KREPES-identified landmarks causes orders-of-magnitude larger confidence drops than random deletion. This validates our analytical framework successfully isolates the true structural anchors of the SSL latent space, rather than identifying geometrical artifacts.

## 6. Conclusion

We introduced KREPES, a unified framework that bridges generalization of SSL with transparency of kernel methods by leveraging eNTK surrogate and Nyström approximation. We resolve the scalability bottleneck of kernels, enabling interpretability of models pretrained on massive datasets via "Representer Landmarks".

## Acknowledgements

This work has been supported by the German Research Foundation (DFG) through the Research Grant GH 257/4-1, and by the DAAD program Konrad Zuse Schools of Excellence in Artificial Intelligence, sponsored by the Federal Ministry of Education and Research (BMBF). This research was also supported by Munich Center for Machine learning (MCML).

## Impact Statement

This paper presents work whose goal is to advance the field of interpretability and transparency in machine learning. There are potential societal consequences of our work, none of which we feel must be specifically highlighted here.

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

## A. Efficient Empirical NTK Calculation

We adopt eNTKs instantiated from a wide range of architectures, including MLPs, CNNs with pooling, ResNets (18, 34, 50), and self-attention blocks.

For a network with parameters $\theta \in \mathbb{R}^P$ and $C$ output classes, let $f(x; \theta) \in \mathbb{R}^C$, $\phi_{\text{eNTK}}(x) = \frac{\partial f(x;\theta)}{\partial \theta} \in \mathbb{R}^{P \times C}$. The eNTK between inputs $x_i, x_j$ is defined by the Jacobian inner product $K_{\text{eNTK}}(x_i, x_j) = \phi_{\text{eNTK}}(x_i)^\top \phi_{\text{eNTK}}(x_j) \in \mathbb{R}^{(N \cdot C) \times (N \cdot C)}$. When the output layer is randomly initialized, the expected kernel factorizes as $\mathbb{E}[K_{\text{eNTK}}] \approx I_C \otimes K_0$, ($I_C := C \times C$ identity, $K_0 \in \mathbb{R}^{N \times N}$ is the kernel for a single logit)(Lee et al., 2020). In practice, we approximate $K_{\text{eNTK}} \approx I_C \otimes K$, where $K$ is computed from one randomly initialized output head. To further accelerate computation, we adapt the parallel eNTK method of Wei et al. (2022) and integrate Nyström approximation.

## B. Further Details on KREPES

---

**Algorithm 1** KREPES: Scalable Analytical Kernel Representation Inference

---

**Input:** Unlabeled dataset $\mathcal{X}$, number of landmarks $m$, Output dimension $h$.
**Output:** Learned projection $\tilde{A} \in \mathbb{R}^{m \times h}$, bias $\gamma \in \mathbb{R}^h$.

**// Phase 1: Nyström Initialization**
Sample landmarks $\mathcal{Z} = \{\tilde{x}_1, \ldots, \tilde{x}_m\} \subset \mathcal{X}$ via K-Means++ or Leverage Score Sampling.
Compute $K_{mm} \approx U_h \Lambda_h U_h^\top$.
Initialize $\tilde{A}_0 \leftarrow U_h \Lambda_h^{-1/2}$ and $\gamma_0 \leftarrow \mathbf{0}$.

**// Phase 2: Batch-wise Gradient Accumulation**
Initialize global gradient $g \leftarrow \mathbf{0}_{m \times h}$.
**for** each batch $X_b \subset \mathcal{X}$ **do**
    Compute cross-view kernel matrices $K_{nm}^A, K_{nm}^B$ against landmarks $\mathcal{Z}$.
    *Forward Pass:* $Z_A \leftarrow K_{nm}^A \tilde{A}_0 + \gamma_0, \quad Z_B \leftarrow K_{nm}^B \tilde{A}_0 + \gamma_0$.
    $g \leftarrow g + \nabla_{\tilde{A}_0} \mathcal{L}_b(Z_A, Z_B)$ {Accumulate exact global gradient}
**end for**

**// Phase 3: Analytical Single-Step Solve (GGN)**
Define dynamic batch-wise HVP function for an arbitrary vector $v$:
    $HVP_{\text{global}}(v) = \sum_{\text{batches}} \text{vjp}\left(\text{res}_b, \tilde{A}_0, \text{jvp}(\text{res}_b, \tilde{A}_0, v)\right) + \lambda v$
$\Delta \tilde{A} \leftarrow \text{ConjugateGradient}\left(HVP_{\text{global}}, -g, \text{Diagonal}(HVP_{\text{global}})\right)$

**// Phase 4: Deterministic Update**
$\tilde{A} \leftarrow \tilde{A}_0 + \Delta \tilde{A}$
$\gamma \leftarrow \gamma_0 + \Delta \gamma$ {Computed concurrently}
**Return** $\tilde{A}, \gamma$

---

Algorithm 1 depicts KREPES's procedure for obtaining global influence vectors.

### B.1. Example of Nyström for KPCA

The kernelized reconstruction loss with Nyström approximation becomes

$$
\frac{1}{n} \sum_{i=1}^{n} \left\| \Phi(x_i) - WW^\top \Phi(x_i) \right\|_{\mathcal{H}}^2 + \lambda \|W\|_{\mathcal{H}}^2
$$
$$
= \frac{\text{Tr}(K)}{n} - \frac{1}{n} \text{Tr}\left( \tilde{A}\tilde{A}^\top K_{nm}^\top K_{nm} \right) + \frac{1}{n} \text{Tr}\left( \tilde{A}\tilde{A}^\top K_{mm} \tilde{A}\tilde{A}^\top K_{nm}^\top K_{nm} \right) + \lambda \, \text{Tr}\left( \tilde{A}^\top K_{mm} \tilde{A} \right).
$$

**B.2. Example of Nyström for Barlow Twins**

To define the kernelized SSL objective for Barlow Twins, we first define embeddings as $Z_A = \tilde{A}K_A + \gamma_A$, $Z_B = \tilde{A}K_B + \gamma_B$, where $K_A$ (resp. $K_B$) are kernels between anchor (resp. positive) samples and landmarks. BT minimizes redundancy via the cross-correlation matrix $C$ defined element-wise as $C_{ij} = \dfrac{\sum_{b=1}^{n} z_{b,i}^A z_{b,j}^B}{\sqrt{\sum_{b=1}^{n}(z_{b,i}^A)^2}\sqrt{\sum_{b=1}^{n}(z_{b,j}^B)^2}}$, where in kernel setting

$$z_{b,i}^A = \sum_{l=1}^{m} \tilde{A}_{il} k_A(x_l, x_b) + \gamma_i, \quad z_{b,j}^B = \sum_{l'=1}^{m} \tilde{A}_{jl'} k_B(x_{l'}, x_b) + \gamma_j$$ ($l$ indexes landmarks from anchor samples and $l'$ those from positive views).

## C. Downstream Classification Accuracy

For performance evaluation, we report downstream classification test accuracy. We first analytically infer the KREPES representations from the unsupervised data manifold without label supervision. We then train a linear classifier on these extracted representations, utilizing labels solely for the final classifier training.

### C.1. Data Pre-processing and Augmentation

For all datasets, we ensure proper feature normalization. In tabular datasets, categorical columns are encoded numerically, while numerical features are standardized. To enhance generalization, we apply data augmentation by adding Gaussian noise and randomly dropping features. In some cases we use the tabular augmentation pipeline in Bahri et al. (2022), which showed small improvements in a few cases.

For CIFAR-10 and ImageNet we normalize using dataset's mean and standard variation. For augmentation we follow the asymmetric augmentation pipeline implemented in solo-learn library da Costa et al. (2022).

### C.2. Baseline Architecture and the Empirical NTK

In all of the metrics corresponding to validating surrogate faithfulness in Table 5. For cifar10 and ImageNet, we used hyper-parameter settings and SSL pretrained checkpoints (corresponded to any SSL loss) provided by da Costa et al. (2022), where they used ResNet-18 for CIFAR-10 and a ResNet-50 for ImageNet. For CIFAR-10 KREPES has ($2 \times m \times h$) $500k/1M$ parameters $m$ being the number of landmarks setting to $1000$ and $h$ depending on the loss function being $250/512$. For ImageNet KREPES has $m = 2000, h = 250/1024$ in it's best performance therefore having $1M/4M$ trainable parameters. For MNIST we used a small 3-layer Relu MLP. For FairFace we use a pretrained ResNet-34.

For all the tabular datasets (except Bank for which we only used a the same MLP as for MNIST, we use the following architecture as the neural network in KREPES:

**Description.** We use a residual MLP backbone with integrated self-attention, which we refer to as *ResMLP*. The input $x \in \mathbb{R}^d$ is first projected to width $w$ using a linear layer followed by GELU activation. A residual self-attention block (LayerNorm + single-head self-attention + skip connection) is then applied. The model further includes $B$ bottleneck MLP blocks, each consisting of a two-layer feedforward network with hidden dimension $w/r$ (where $r$ is the bottleneck factor), residual connection, and GELU activation. Finally, the representation is projected to the output dimension. Unless otherwise stated, we use $w = 1032$, $B = 3$, $r = 4$, and single-head attention in all experiments. The full PyTorch implementation is provided in the supplementary material.

The feature dimension d varies across datasets, influencing the total number of parameters: $d = 110$ for Adult, $d = 54$ for CoverType, and $d = 28$ for Higgs. Nonetheless, all resulting neural networks have roughly 5 million parameters. The total number of KREPES parameters ($2 \times m \times h$) depends on the number of landmarks and the representation dimension $h$. For all datasets we use $m = 1000$, yielding between $500k$ and $900k$ KREPES parameters depending on h.

## D. Implemented Representation Learning Objectives

We categorize popular losses for self/un-supervised representation learning into: 1. reconstruction-based ($p = 1$), 2. contrastive, 3. joint embedding, and 4. predictive self-distillation ($p \geq 2$). The followings elaborate on these categories.

---

**Algorithm 2** ResMLP Forward Pass

---

1: **Input:** $x \in \mathbb{R}^d$, width $w$, bottleneck factor $r$, number of blocks $B$
2: $x \leftarrow \text{Linear}(d \rightarrow w)(x)$
3: $x \leftarrow \text{GELU}(x)$
4: $x \leftarrow x + \text{SelfAttention}(\text{LayerNorm}(x))$
5: **for** $i = 1$ **to** $B$ **do**
6:     $h \leftarrow \text{Linear}(w \rightarrow w/r)(x)$
7:     $h \leftarrow \text{GELU}(h)$
8:     $h \leftarrow \text{Linear}(w/r \rightarrow w)(h)$
9:     $x \leftarrow \text{GELU}(x + h)$
10: **end for**
11: **Return** $x$

---

**Joint Embedding Objectives** $\mathcal{L}(Z_A, Z_B)$, the objective is to align embeddings $Z_A$ and $Z_B$, obtained from two views of the same data sample. We implement the kernelized version of two famous losses of this categories; Barlow Twins (BT) which minimizes redundancy between dimensions of learned embeddings by regularizing their cross-correlation matrix, and VICReg (Variance-Invariance-Covariance Regularization) which is designed to balance the variance, invariance, and covariance between the embeddings.

**Predictive Self-distillation Objectives** We implement kernel BYOL (Grill et al., 2020b) loss function with the same definition.

**Contrastive Objectives** $\{x^p\} \equiv \{x, x^+, x^-\}$, denoting the anchor, the positive and the negative examples, respectively. We use the same definition as Esser et al. (2024) for the simple and spectral contrastive loss plus the aforementioned Tikhonov regularization term as the introduced orthogonality constraint. For SimCLR loss function we used the same exact definition as Chen et al. (2020).

**Reconstruction-based Objectives** KPCA generalizes PCA to non-linear feature space by mapping into RKHS $\mathcal{H}$ via feature map $\Phi$. Principal components in $\mathcal{H}$ are identified by maximizing the variance of the projected data while minimizing reconstruction loss. Kernel Auto-Encoder (KAE), minimizes a reconstruction loss with RKHS regularization as defined in Esser et al. (2024).

## E. Influence Functions in Self-Supervised Learning

Let $P$ be a distribution on a measurable space $(\mathcal{X}, \mathcal{A})$ and let $T$ be a statistical functional taking values in $\mathbb{R}^p$. For $z \in \mathcal{X}$, define the contaminated distribution

$$P_{\epsilon,z} := (1 - \epsilon)P + \epsilon\delta_z, \quad \epsilon \in [0, 1], \tag{16}$$

where $\delta_z$ is the Dirac measure at $z$. The influence function of $T$ at $P$ in direction $z$ is defined as

$$\text{IF}(z; T, P) := \left. \frac{d}{d\epsilon} T(P_{\epsilon,z}) \right|_{\epsilon=0}. \tag{17}$$

This corresponds to the Gâteaux derivative of $T$ at $P$ (Hampel et al., 1986; van der Vaart, 1998).

**Implicit Function Theorem.** Let $F : \mathbb{R}^p \times \mathcal{P} \rightarrow \mathbb{R}^p$. Assume that $F$ is continuously differentiable in $\theta$ in a neighborhood of $\theta(P)$, that $J_\theta(P) := \nabla_\theta F(\theta(P), P)$ is invertible, and that the directional derivative $\left. \frac{d}{d\epsilon} F(\theta(P), P_{\epsilon,z}) \right|_{\epsilon=0}$ exists. Then, the implicit function theorem (Rudin, 1976) gives a locally differentiable path $\epsilon \mapsto \theta(P_{\epsilon,z})$ satisfying $F(\theta(P_{\epsilon,z}), P_{\epsilon,z}) = 0$, and

$$\text{IF}(z; \theta, P) = -J_\theta(P)^{-1} \left. \frac{d}{d\epsilon} F(\theta(P), P_{\epsilon,z}) \right|_{\epsilon=0}. \tag{18}$$

**Example: Barlow Twins.** A standard supervised population risk has the form

$$\mathcal{L}(P, \theta) = \mathbb{E}_P[\ell(X, \theta)]. \tag{19}$$

The Barlow Twins objective is not, in general, a pointwise additive risk of the form $\mathbb{E}_P[\ell(X, \theta)]$. Instead, its population loss is a differentiable function of expectation-type statistics of the data distribution, such as the population cross-correlation matrix.

Let $X \sim P$ and let $V_1, V_2$ be augmentations. Let $f_\theta$ be the encoder and define $u_\theta(X, V_1) := f_\theta(V_1(X))$, $\quad v_\theta(X, V_2) := f_\theta(V_2(X))$. Define $g_\theta(X, V_1, V_2) := u_\theta(X, V_1)v_\theta(X, V_2)^\top$. If $S = (X, V_1, V_2)$ and $Q_P$ denotes the distribution of $S$ induced by $X \sim P$ and the fixed augmentation kernels, then the population cross-correlation matrix is

$$C(P, \theta) = \mathbb{E}_{S \sim Q_P}[g_\theta(S)]. \tag{20}$$

Thus the Barlow Twins population objective can be written as

$$\mathcal{L}_{BT}(P, \theta) = \Phi(C(P, \theta)), \tag{21}$$

where

$$\Phi(C) = \sum_i (C_{ii} - 1)^2 + \lambda \sum_{i \neq j} C_{ij}^2. \tag{22}$$

Therefore, Barlow Twins is compatible with our definition of influence-function because

$$P \mapsto C(P, \theta) = \mathbb{E}_{Q_P}[g_\theta(S)] \mapsto \Phi(C(P, \theta)) \tag{23}$$

is a smooth functional of an expectation. Define

$$F(\theta, P) := \nabla_\theta \mathcal{L}_{BT}(P, \theta), \tag{24}$$

under the optimality condition $F(\theta(P), P) = 0$. Let $A(C) := \nabla_C \Phi(C)$, with entries

$$A(C)_{ij} = \begin{cases} 2(C_{ii} - 1), & i = j, \\ 2\lambda C_{ij}, & i \neq j. \end{cases} \tag{25}$$

By the chain rule,

$$F(\theta, P) = \nabla_\theta \mathcal{L}_{BT}(P, \theta) = (\nabla_\theta C(P, \theta))^\top A(C(P, \theta)). \tag{26}$$

*(Note: Here, matrices are understood as vectorized when multiplied; equivalently, the above expression uses the Frobenius inner product.)*

Assume that the augmentation kernels do not depend on $P$. Contamination of $P$ induces contamination of the joint distribution $Q_P$

$$Q_{P_{\epsilon,z}} = (1 - \epsilon)Q_P + \epsilon Q_z, \tag{27}$$

where $Q_z$ is the distribution obtained by fixing the base sample to $z$ and drawing augmentations $V_1, V_2$ from their usual kernels. Hence,

$$C(P_{\epsilon,z}, \theta) = \mathbb{E}_{S \sim Q_{P_{\epsilon,z}}}[g_\theta(S)] \tag{28}$$

$$= (1 - \epsilon)\mathbb{E}_{S \sim Q_P}[g_\theta(S)] + \epsilon \mathbb{E}_{S \sim Q_z}[g_\theta(S)]. \tag{29}$$

Define $C_z(\theta) := \mathbb{E}_{V_1, V_2}\left[f_\theta(V_1(z))f_\theta(V_2(z))^\top\right]$. Then $C(P_{\epsilon,z}, \theta) = (1 - \epsilon)C(P, \theta) + \epsilon C_z(\theta)$, and hence

$$\left.\frac{d}{d\epsilon}C(P_{\epsilon,z}, \theta)\right|_{\epsilon=0} = C_z(\theta) - C(P, \theta). \tag{30}$$

We treat a realized augmented pair as the sample, then $C_z(\theta)$ reduces to $g_\theta(z) = u_\theta(z)v_\theta(z)^\top$.

Furthermore, let $\nabla_\theta C(P, \theta) = \mathbb{E}_{S \sim Q_P}[\nabla_\theta g_\theta(S)]$. For example, when $g_\theta(S)$ is differentiable in $\theta$ and there exists an integrable envelope $M(S)$ such that, locally around $\theta$, $\|\nabla_\theta g_\theta(S)\| \leq M(S)$. Then under contamination,

$$\nabla_\theta C(P_{\epsilon,z}, \theta) = (1 - \epsilon)\nabla_\theta C(P, \theta) + \epsilon \nabla_\theta C_z(\theta), \tag{31}$$

Hence,

$$\frac{d}{d\epsilon}\nabla_\theta C(P_{\epsilon,z},\theta)\bigg|_{\epsilon=0} = \nabla_\theta C_z(\theta) - \nabla_\theta C(P,\theta). \tag{32}$$

This step does not require $\mathcal{L}_{BT}$ to be pointwise additive. It only requires that $C(P,\theta)$ and $\nabla_\theta C(P,\theta)$ are expectations under the induced augmentation distribution.

Let $C := C(P,\theta(P))$. At fixed $\theta = \theta(P)$, we differentiate

$$F(\theta(P), P_{\epsilon,z}) = (\nabla_\theta C(P_{\epsilon,z},\theta(P)))^\top A(C(P_{\epsilon,z},\theta(P))). \tag{33}$$

Using the product rule,

$$\frac{d}{d\epsilon}F(\theta(P), P_{\epsilon,z})\bigg|_{\epsilon=0} = \left(\frac{d}{d\epsilon}\nabla_\theta C(P_{\epsilon,z},\theta(P))\bigg|_{\epsilon=0}\right)^\top A(C) + (\nabla_\theta C(P,\theta(P)))^\top \frac{d}{d\epsilon}A(C(P_{\epsilon,z},\theta(P)))\bigg|_{\epsilon=0}$$
$$= (\nabla_\theta C_z(\theta(P)) - \nabla_\theta C(P,\theta(P)))^\top A(C) + (\nabla_\theta C(P,\theta(P)))^\top \nabla_C A(C) (C_z(\theta(P)) - C). \tag{34}$$

Finally, let $H := \nabla_\theta^2 \mathcal{L}_{BT}(P, \theta(P))$. Assuming $H$ is invertible,

$$\text{IF}(z;\theta,P) = -H^{-1}\left[(\nabla_\theta C_z(\theta(P)) - \nabla_\theta C(P,\theta(P)))^\top A(C) + (\nabla_\theta C(P,\theta(P)))^\top \nabla_C A(C) (C_z(\theta(P)) - C)\right]. \tag{35}$$

**Vectorized form.** The derivation above uses matrix notation for readability. Equivalently, one may write the same expression in vectorized form. Let $c(P,\theta) = \text{vec}(C(P,\theta))$, $a(c) = \nabla_c \Phi(c)$, so that

$$F(\theta, P) = (\nabla_\theta c(P,\theta))^\top a(c(P,\theta)).$$

Let $W := \nabla_c a(c) = \nabla_c^2 \Phi(c)$. For Barlow Twins, $W$ is diagonal with entries 2 on diagonal correlation terms and $2\lambda$ on off-diagonal terms. Defining $c_z(\theta) = \text{vec}(C_z(\theta))$, then

$$\frac{d}{d\epsilon}F(\theta(P), P_{\epsilon,z})\bigg|_{\epsilon=0} = (\nabla_\theta c_z(\theta) - \nabla_\theta c(P,\theta))^\top a(c) + (\nabla_\theta c(P,\theta))^\top W (c_z(\theta) - c(P,\theta)).$$

Thus the influence function can be written compactly as

$$\text{IF}(z;\theta,P) = -H^{-1}\frac{d}{d\epsilon}F(\theta(P), P_{\epsilon,z})\bigg|_{\epsilon=0}, \qquad H = \nabla_\theta^2 \mathcal{L}_{BT}(P, \theta(P)).$$

**Remark on realized augmentations.** If the sample is taken to be a realized augmented pair rather than a base image, then $C_z(\theta)$ is replaced by $g_\theta(z)$ and $\nabla_\theta C_z(\theta)$ is replaced by $\nabla_\theta g_\theta(z)$.

In that case the expression becomes

$$\text{IF}(z;\theta,P) = -H^{-1}\left[(\nabla_\theta g_\theta(z) - \mathbb{E}[\nabla_\theta g_\theta(S)])^\top A(C) + (\nabla_\theta C)^\top \nabla_C A(C) (g_\theta(z) - C)\right]. \tag{36}$$

The augmentation-averaged version above is the appropriate form when contamination is applied to the underlying data distribution $P$ over base samples.

**Connection to the Finite Dimensional KREPES's Score.** The expression above is the population influence function for the minimizer $\theta(P)$. In KREPES implementation, $\theta$ is replaced by the finite-dimensional Nyström coefficient matrix $\tilde{A}$, and the exact Hessian is replaced by the regularized GGN curvature $\bar{H}_{GN}$. Consequently, the population influence displacement $-H^{-1}\frac{d}{d\epsilon}F_z(P)$ is instantiated by the deterministic local step

$$\text{vec}(\Delta A) = -\bar{H}_{GN}^{-1}\text{vec}(\nabla_A \mathcal{L}(A_0)),$$

as used in (5). The landmark-wise score in (6) is then the exact pushforward of this parameter displacement through the representer map $f(x_t; A) = A^\top k_{x_t} + \gamma$. Thus Appendix E justifies the influence-function form, while Appendix F justifies the PD curvature surrogate used to make the step well-defined for non-convex SSL losses.

# F. Derivations of Generalized Gauss-Newton Hessian Approximation

Let the kernel matrix be $K \in \mathbb{R}^{n \times m}$, and the learned parameter matrix be $\tilde{A} \in \mathbb{R}^{m \times h}$. The representations are given by the encoder function $f_\theta$

$$\mathbf{Z} = \mathbf{K}\tilde{\mathbf{A}} + \gamma \quad \in \mathbb{R}^{n \times h}$$

where $\theta = \text{vec}(\tilde{\mathbf{A}})$ denotes the flattened parameters. For the derivation, we assume $\gamma$ is absorbed or negligible for the curvature of $\tilde{\mathbf{A}}$. We define the Jacobian of the representations with respect to the parameters as $\mathbf{J}_Z = \frac{\partial \text{vec}(\mathbf{Z})}{\partial \theta}$.

## F.1. Barlow Twins (BT)

We treat the Barlow Twins objective as a Non-Linear Least Squares problem. This allows us to utilize the classic Gauss-Newton approximation, which guarantees a Positive Semi-Definite (PSD) curvature matrix. The original loss is defined on the cross-correlation matrix $\mathcal{C} \in \mathbb{R}^{h \times h}$ between normalized batches $\hat{\mathbf{Z}}_A$ and $\hat{\mathbf{Z}}_B$

$$\mathcal{L}_{BT} = \sum_i (1 - \mathcal{C}_{ii})^2 + \lambda_{reg} \sum_{i \neq j} \mathcal{C}_{ij}^2$$

We formally reframe this as the squared $L_2$ norm of a **residual vector** $r(\theta)$. Let $\mathbf{W} \in \mathbb{R}^{h \times h}$ be a weighting matrix, where $W_{ii} = 1$ and $W_{ij} = \sqrt{\lambda_{reg}}$ for $i \neq j$. We define the residual map $r : \mathbb{R}^{m \times h} \to \mathbb{R}^{h^2}$ as

$$r(\theta) = \text{vec}\left(\mathbf{W} \odot (\mathcal{C}(\theta) - \mathbf{I})\right)$$

The loss becomes

$$\mathcal{L}_{BT}(\theta) = \|r(\theta)\|_2^2$$

**The Generalized Gauss-Newton Approximation** The exact Hessian of such a loss is $\nabla^2 L = 2J_r^\top J_r + 2\sum_i r_i \nabla^2 r_i$. The Gauss-Newton method discards the second-order term (which involves the Hessian of the residuals), assuming the residuals are small or the function is locally linear. The approximate Hessian is

$$\mathbf{H}_{GN}^{BT} = 2\mathbf{J}_r^\top \mathbf{J}_r$$

where $\mathbf{J}_r = \frac{\partial r}{\partial \theta} \in \mathbb{R}^{h^2 \times mh}$ is the Jacobian of the residual vector with respect to the parameters.

However, explicitly forming $\mathbf{J}_r^\top \mathbf{J}_r$ is computationally prohibitive ($O((mh)^2)$). We compute the Hessian-Vector Product (HVP) for an arbitrary direction vector $v \in \mathbb{R}^{m \times h}$ using automatic differentiation

$$\mathbf{H}_{GN}^{BT} v = 2\mathbf{J}_r^\top (\mathbf{J}_r v)$$

We implement this as a two-step procedure

1. JVP (Forward Mode): Computing the directional derivative of the residuals

$$u = \mathbf{J}_r v = \text{jvp}(r, \theta, v)$$

2. VJP (Reverse Mode): Propagating the result back through the transpose Jacobian

$$\mathbf{H}_{GN}^{BT} v = 2 \cdot \text{vjp}(r, \theta, u)$$

## F.2. SimCLR (SC)

For SimCLR, the loss is not a sum of squares but a Cross-Entropy (CE) loss. The standard Gauss-Newton approximation for CE is defined by the decomposition of the Hessian into the structure $\mathbf{J}^\top \mathbf{Q} \mathbf{J}$, where $\mathbf{Q}$ is the Hessian of the CE loss with respect to the logits.

Let $\ell(\theta) \in \mathbb{R}^{2n \times 2n}$ be the matrix of pairwise similarities (logits), scaled by temperature $\tau$ ($\ell_{i,i}$ is masked before the softmax). The loss is the row-wise cross-entropy

$$\mathcal{L}_{SC} = \sum_{i=1}^{2n} \mathrm{CE}(p_i, y_i) = - \sum_{i=1}^{2n} \log p_{i,y_i}.$$

where $\mathbf{p}_i = \mathrm{softmax}(\ell_{i,:})$ is the probability vector for sample $i$.

**The Generalized Gauss-Newton Approximation**    Let $\phi$ map parameters to logits $l = \phi(\theta)$, and let $L$ be the loss composed with $\phi$. The exact Hessian is

$$J_\phi^\top \, \nabla_\ell^2 L \, J_\phi \; + \; \sum_h (\nabla_\ell L)_h \, \nabla_\theta^2 \phi_h.$$

The GGN approximation drops the second term, retaining the curvature of the loss function

$$\mathbf{H}_{GN}^{SC} = \mathbf{J}\phi^\top \mathbf{Q} \mathbf{J}_\phi$$

Where $Q = \nabla_l^2 L$ is a block-diagonal matrix and each block corresponds to the Hessian of the softmax-cross-entropy for a single row $i$ with probability vector $\mathbf{p}_i$. The corresponding Hessian block is

$$\mathbf{Q}_i = \mathrm{diag}(\mathbf{p}_i) - \mathbf{p}_i \mathbf{p}_i^\top$$

To apply the preconditioner $\mathbf{H}_{GN}^{SC} v = \mathbf{J}_\phi^\top \mathbf{Q} \mathbf{J}_\phi v$, we utilize the chain rule structure inherent in the GGN

1. JVP (Forward Mode): Computing the perturbation in logits given perturbation in parameters $v$

$$\delta \ell = \mathbf{J}_\phi v = \mathrm{jvp}(\phi, \theta, v)$$

2. $\mathbf{Q}$ (Softmax Geometry): Applying the Hessian of the CE loss to the logit perturbation. This operation is efficient because $\mathbf{Q}$ is never materialized; we compute the matrix-vector product row-wise

$$\delta g_i = \mathbf{Q}_i(\delta \ell_i) = \mathbf{p}_i \odot (\delta \ell_i) - \mathbf{p}_i(\mathbf{p}_i^\top \delta \ell_i)$$

where $\delta g$ represents the resulting gradient perturbation.

3. VJP (Reverse Mode): Backpropagating the gradient perturbation to parameter space

$$\mathbf{H}_{GN}^{SC} v = \mathbf{J}_\phi^\top (\delta g) = \mathrm{vjp}(\phi, \theta, \delta g)$$

## F.3. VICReg

VICReg is structurally more intricate than Barlow Twins as it consists of three distinct terms: Invariance, Variance, and Covariance. Let $\mathbf{Z}_A, \mathbf{Z}_B \in \mathbb{R}^{n \times h}$ be the batch embeddings. The objective is:

$$\mathcal{L}_{VIC} = \lambda \mathcal{L}_{inv} + \mu \mathcal{L}_{var} + \nu \mathcal{L}_{cov}$$

Where the individual components are defined as

- **Invariance:** $\mathcal{L}_{inv} = \frac{1}{n} \|\mathbf{Z}_A - \mathbf{Z}_B\|_F^2$

- **Covariance:** $\mathcal{L}_{cov} = \frac{1}{h}\sum_{i\neq j}[\mathcal{C}(\mathbf{Z}_A)]_{i,j}^2 + \frac{1}{h}\sum_{i\neq j}[\mathcal{C}(\mathbf{Z}_B)]_{i,j}^2$, where $\mathcal{C}(\mathbf{Z})$ is the $h \times h$ covariance matrix.

- **Variance:** $\mathcal{L}_{var} = \frac{1}{h}\sum_{j=1}^{h}\max(0, \gamma - S(\mathbf{Z}_{A,\cdot j})) + \frac{1}{h}\sum_{j=1}^{h}\max(0, \gamma - S(\mathbf{Z}_{B,\cdot j}))$, where $S$ is the standard deviation across the batch.

To apply the GGN approximation ($\mathbf{H}_{GN} = \mathbf{J}^\top \mathbf{Q}\mathbf{J}$), we recast the objective as a Non-Linear Least Squares (NLLS) problem. For the variance term ($\mathcal{L}_{var}$) we treat the hinge penalty as the outer loss in the GGN decomposition. Away from the kink, a piecewise-linear hinge has zero second derivative with respect to its input, and it is non-differentiable at the kink. We therefore omit this term from the GGN curvature ($\mathbf{Q}$) and retain its contribution in the first-order gradient. This does not imply that the exact parameter Hessian of the variance term is zero; it only specifies the curvature approximation used in our GGN surrogate.

**Formulating the NLLS Residuals.** Since the variance curvature is zero, we construct the residual vector $r(\theta)$ solely for the Invariance and Covariance terms. We scale these residuals such that $\frac{1}{2}\|r(\theta)\|_2^2$ perfectly recovers the weighted loss components. For the invariance term, to satisfy $\frac{1}{2}\|r_{inv}\|_2^2 = \lambda\mathcal{L}_{inv}$, we define

$$r_{inv}(\theta) = \sqrt{\frac{2\lambda}{n}}\mathrm{vec}(\mathbf{Z}_A - \mathbf{Z}_B)$$

Let offdiag($\mathbf{M}$) be an operation that flattens a matrix into a vector while discarding the diagonal elements. To satisfy $\frac{1}{2}\|r_{cov}\|_2^2 = \nu\mathcal{L}_{cov}$, we define

$$r_{cov,A}(\theta) = \sqrt{\frac{2\nu}{h}}\mathrm{offdiag}(\mathcal{C}(\mathbf{Z}_A))$$

$$r_{cov,B}(\theta) = \sqrt{\frac{2\nu}{h}}\mathrm{offdiag}(\mathcal{C}(\mathbf{Z}_B))$$

We concatenate these into a single unified residual mapping $r : \mathbb{R}^{m\times h} \to \mathbb{R}^{nh+2(h^2-h)}$:

$$r(\theta) = \begin{bmatrix} r_{inv}(\theta) \\ r_{cov,A}(\theta) \\ r_{cov,B}(\theta) \end{bmatrix}$$

**The Generalized Gauss-Newton Approximation** With the unified residual vector established as $\frac{1}{2}\|r(\theta)\|_2^2$ minimization problem, the analytical Generalized Gauss-Newton Hessian with respect to the parameters is

$$\mathbf{H}_{GN}^{VIC} = \mathbf{J}_r^\top \mathbf{J}_r$$

where $\mathbf{J}_r = \frac{\partial r}{\partial \theta}$ is the Jacobian of the unified residual vector. Similar to Barlow Twins, we circumvent the explicit construction of $\mathbf{J}_r$ by evaluating the exact Hessian-Vector Product for an arbitrary direction vector $v \in \mathbb{R}^{m\times h}$ dynamically via forward-over-reverse automatic differentiation:

$$\mathbf{H}_{GN}^{VIC}v = \mathbf{J}_r^\top (\mathbf{J}_r v)$$

We implement this using the following two-step procedure:

1. JVP (Forward Mode): Computing the directional derivative of the concatenated residuals

$$u = \mathbf{J}_r v = \mathrm{jvp}(r, \theta, v)$$

2. VJP (Reverse Mode): Propagating the result back through the transpose Jacobian

$$\mathbf{H}_{GN}^{VIC}v = \mathrm{vjp}(r, \theta, u)$$

---

**Algorithm 3** Principal Component Initialization (PCI)

---

1: **Function** PCI($K_{mm} \in \mathbb{R}^{m \times m}, h, \tilde{A}_0 \in \mathbb{R}^{m \times h}$)
2: *// Considering Nyström approximation $K_{nn} \approx K_{nm} K_{mm}^\dagger K_{mn}$*
3: Rank-h eigendecomposition $K_{mm} \approx U_h \Lambda_h U_h^\top, U_h \in \mathbb{R}^{m \times h}, \Lambda_h \in \mathbb{R}^{h \times h}$ then $K_{mm}^\dagger \approx U_h \Lambda_h^{-1} U_h^\top$
4: *// Define $\Phi = K_{nm} U_h \Lambda_h^{-1/2} \in \mathbb{R}^{n \times h}$, s.t. $K_{nn} \approx \Phi \Phi^\top$*
5: Compute the initial parameter matrix
6: $\tilde{A}_0 \leftarrow U_h (\Lambda_h + \epsilon \mathbf{I}_h)^{-1/2}$
7: **Return** $\tilde{A}_0$

---

### F.4. BYOL

BYOL compares the online prediction of one augmented view to the stop-gradient target representation of the other view. Let $z_A^a = f_A(x^a)$, $z_A^b = f_A(x^b)$, and let $q$ denote the predictor, $p_A^a = q(z_A^a)$, $p_A^b = q(z_A^b)$, and normalized vectors $\hat{p}_A^a = \frac{p_A^a}{\|p_A^a\|_2}$, $\hat{z}_A^a = \frac{z_A^a}{\|z_A^a\|_2}$, with analogous definitions for view $b$. The stop-gradient BYOL residuals are

$$r_{a \to b}(A) = \hat{p}_A^a - \text{sg}(\hat{z}_A^b), \qquad r_{b \to a}(A) = \hat{p}_A^b - \text{sg}(\hat{z}_A^a).$$

Concatenating the two directions,

$$r_{BYOL}(A) = \begin{bmatrix} r_{a \to b}(A) \\ r_{b \to a}(A) \end{bmatrix},$$

the BYOL loss is written as the NLLS objective

$$\mathcal{L}_{BYOL}(A) = \|r_{BYOL}(A)\|_2^2.$$

Therefore the Gauss–Newton curvature is

$$H_{GN}^{BYOL} = 2 J_{r_{BYOL}}^\top J_{r_{BYOL}},$$

where $J_{r_{BYOL}} = \frac{\partial r_{BYOL}}{\partial \theta}$. The corresponding HVP for a direction $d$ is

$$\text{HVP}_{BYOL}(d) = 2 \cdot \text{vjp}(r_{BYOL}, A, \text{jvp}(r_{BYOL}, A, d)).$$

## G. Nyström Initialization

Algorithm 3 shows the process of initializing the parameters discussed in 4.1.

**Ablation study on Parameter initialization.** In Table 3 we report the ablation study results regarding our proposed Principal Component Initialization method and the random Kaiming initialization from He et al. (2015).

*Table 3.* Downstream classification test accuracy across three datasets and three losses with random Kaiming initialization and our proposed Principal Component Initialization (PCI) method.

| Dataset | BT | | SimCLR | | BYOL | |
|---|---|---|---|---|---|---|
| | Kaiming | PCI | Kaiming | PCI | Kaiming | PCI |
| Adult-1M | 83.49 | 84.05 | 83.15 | 83.67 | 83.34 | 83.76 |
| CoverType | 83.48 | 85.82 | 81.00 | 85.16 | 82.94 | 85.31 |
| CIFAR-10 | 90.01 | 91.18 | 90.12 | 90.36 | 89.85 | 90.56 |

## H. Landmark Selection Methods

Since Nyström is a low-rank approximation, the landmarks $\mathcal{Z} = \{\tilde{x}_l\}_{l=1}^m$ define the column space of $K_{nm}$ and the subspace for $\tilde{A}_0$ ( 4.1); hence, selecting them carefully ensures the initial projection captures the most informative directions in kernel space, improving convergence and representation quality. We employ two selection strategies:

**(1) K-means++ Seeding.** This algorithm (Arthur & Vassilvitskii, 2007) in which we choose the first landmark $c_1 \sim$ Uniform($\mathcal{X}$), and initialize the set of selected landmarks as $C = \{c_1\}$, then we continue adding to this list iteratively with the probability $P(x_j) = \dfrac{D(x_j)^2}{\sum_{k=1}^{n} D(x_k)^2}$, where $D(x_j)^2 = \min_{c \in C} \|x_j - c\|_2^2$ until $|C| = m$.

**(2) Approximate Leverage Score Sampling.** We propose a randomized diagonal estimation of the q-approximate leverage scores leveraging the Hutchinson's estimator (Hutchinson, 1989). Let $K \in \mathbb{R}^{n \times n}$, where $K_{ij} = \kappa(x_i, x_j)$. Leverage score for the $j$-th point is defined as $l_j(\lambda) = (K(K + \lambda n I)^{-1})_{jj}$ (Rudi et al., 2017; 2018). Since directly computing this inverse would require $O(n^3)$ operations and $O(n^2)$ memory, we propose a randomized algorithm using Hutchinson's estimator to approximate the diagonal of $M = K(K + \lambda n I)^{-1}$. We generate a random sketching matrix $\Pi \in \mathbb{R}^{n \times s}$, (Halko et al., 2011) where each entry $\Pi_{ij}$ is drawn independently from a distribution (e.g., a standard normal or a sparse Rademacher distribution). We define $Z = (K + \lambda n I)^{-1}\Pi, M\Pi = KZ$ then we solve each linear system $(K + \lambda n I)z_j = p_j, \forall j$, via Conjugate Gradients (Hestenes & Stiefel, 1952). As a result, we have the vector of approximated leverage scores $\hat{\ell}(\lambda) = \sum_{\text{cols}}\big(\Pi \odot (M\Pi)\big)$, and choose landmarks with the probability $P(x_j) = \dfrac{\hat{\ell}_j(\lambda)}{\sum_{k=1}^{n} \hat{\ell}_k(\lambda)}$.

**Ablation Study on Different Landmark Selection Methods.** In table 4 we report the downstream performance on the MNIST dataset, across all the proposed loss functions in the paper with three different landmark selection algorithms.

*Table 4.* Downstream classification test accuracy across all the proposed losses on MNIST dataset with three different algorithms for landmark selection.

|  | Random Uniform | Kmeans++ | Leverage Score |
|---|---|---|---|
| BT | 94.95 | 97.41 | 97.95 |
| VICReg | 90.18 | 89.57 | 89.69 |
| BYOL | 79.53 | 84.68 | 85.30 |
| SimCLR | 95.35 | 98.90 | 95.75 |
| Spectral Cont. | 94.22 | 97.99 | 98.09 |
| Simple Cont. | 88.67 | 89.64 | 89.34 |
| KAE | 96.42 | 98.89 | 98.60 |
| KPCA | 89.12 | 97.62 | 97.63 |

# I. Further Details on Interpretability Analysis

**Sample Specific Influence Score** We observe consistent results on CIFAR-10, as illustrated in Figure 10. The top influential landmarks not only align with the test sample's ground-truth class but also demonstrate fine-grained semantic similarity. For instance, the landmarks retrieved for a specific cat image are not merely generic samples from the 'cat' distribution; rather, they share distinct visual attributes (e.g., pose, color, background) with the test instance, confirming that the model captures subtle intra-class variations.

# J. Concept-Conditioned Influence via Concept Activation Vectors

In order to obtain the concept activation vector $v_c$, we follow the procedure in Kim et al. (2018). First we need to gather the concept positive and negative datasets, one with the concept present in them and the other without the concept. For our example in this paper we choose the concept *"Sea"* from the class Sea in the CIFAR-100 dataset. Hence, all the samples in this class which are 500 images, will be the positive examples for the concept Sea. Next we have the negative examples from the domain on which we retrieved the $\tilde{A}$ in KREPES. We pick 200 random images from a diverse set of classes: cat, dog, horse, truck, automobile.

We compute their representations in the learned kernel space as $Z_P = \tilde{A}^\top K_{P,m}^\top$ for the concept positive samples and $Z_N = \tilde{A}^\top K_{N,m}^\top$ for the concept negative samples, where $K_{P,m} \in \mathbb{R}^{|P_c| \times m}$ and $K_{N,m} \in \mathbb{R}^{|N_c| \times m}$ denote the kernel matrices between the concept sets and the learned representers.

Further, we train a linear binary classifier which has to learn the direction that separates "Sea" from "random". For this purpose we use a linear SVC classifier from sklearn library. The concept vector $v_c$ is then simply the normalized weight

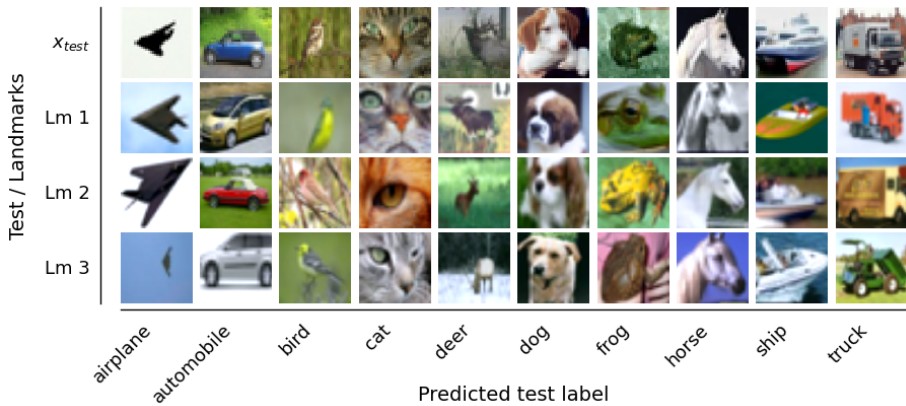

*Figure 10.* Sample-specific influential landmarks. First row displays test samples, followed by their top-3 influential landmarks, ranked by the influence score on CIFAR-10.

---

**Algorithm 4** KREPES Interpretability: Influence Scores and Conceptual Profiles

---

**Input:** Test sample $x_t$, Landmarks $\mathcal{Z} = \{\tilde{x}_l\}_{l=1}^m$, Influence step $\Delta\tilde{A}$, Concept $c$ with CAV $v_c$.
**Output:** Top-$N$ influential landmarks $\mathcal{R}_{\text{top}}$, Conceptual Score $\Psi(x_t; v_c)$.

**// Step 1: Sample-Specific Influence Score**
**for** landmark $l = 1 \ldots m$ **do**
   $\mathcal{IS}(\tilde{x}_l, x_t) \leftarrow \left\| \nabla_{A_{l,:}} f(x_t) \Delta A_{l,:}^\top \right\|_2 = k(x_t, \tilde{x}_l) \left\| \Delta\tilde{A}_{l,:} \right\|_2$ (6)
**end for**
*Identify Top-N landmarks:* $\mathcal{R}_{\text{top}} \leftarrow \text{argsort}\big(\mathcal{IS}(\tilde{x}_l, x_t)\big)[: N]$.

**// Step 2: Concept-Conditioned Influence**
$\Psi(x_t; v_c) \leftarrow 0$
**for** $l \in \mathcal{R}_{\text{top}}$ **do**
   $\mathcal{IS}(\tilde{x}_l, x_t; v_c) \leftarrow \langle \nabla_{A_{l,:}} f(x_t) \Delta A_{l,:}^\top, v_c \rangle$ (8)
   $\Psi(x_t; v_c) \leftarrow \Psi(x_t; v_c) + \mathcal{IS}(\tilde{x}_l, x_t; v_c)$
**end for**
**Return** $\mathcal{R}_{\text{top}}, \Psi(x_t; v_c)$

---

vector learned by the classifier. This vector points from the "random" examples towards the "Sea" examples.

## K. Computational Resources

The experiments are conducted on a high-performance computing (HPC) cluster equipped with NVIDIA H100 and A100 GPUs. For training, we primarily use a single NVIDIA H100 GPU. Depending on the model size, we employed up to four GPUs to compute the eNTKs in parallel. For example, computing the eNTK for ResNet-18 on CIFAR-10 using 2 GPUs takes less than 1 hour.

**LLM Usage.** In this paper, LLMs are used to assist with writing refinement, code debugging, and the search for related work and references. All generated suggestions and outputs are carefully reviewed and verified by the authors.

