# OpenReview forum: "Interpretable Self-Supervised Learning via Representer Landmarks and  Nyström Approximation"
_ICML.cc/2026/Conference — ICML 2026 regular_

### Official Review · Reviewer_LQUW · 2026-03-10

**Soundness:** 2
**Presentation:** 2
**Significance:** 2
**Originality:** 3
**Overall Recommendation:** 4
**Confidence:** 4

**Summary:**

This paper proposes a framework called KREPES, aiming to learn interpretable representations for SSL. This method combines the empirical tangent kernel (eNTK) approximation of neural networks with the representation theorem of kernel methods, directly expressing the learned latent space through representation landmarks. KREPES introduces an optimization strategy based on the Nyström approximation and designs principal component initialization and generalized Gauss-Newton (GGN) preconditioners for non-convex SSL objectives. The authors propose two new metrics and validate the effectiveness of the proposed method.

**Compliance With Llm Reviewing Policy:**

Affirmed.

**Final Justification:**

I have no other concerns. Thus, I increase my rating. However, this paper needs major revision. Thus, if the ACs want to reject this paper, I am ok.

**Key Questions For Authors:**

Please see weaknesses for details.

**Limitations:**

This paper didn't discuss the limitations, and the practical limitations may lie in (i) its central interpretability promise depends heavily on the faithfulness of the eNTK-based proxy to the original SSL neural network, while the current validation is still mostly based on downstream performance rather than explanation-level agreement; and (ii) although the training objective is self-supervised, several of the interpretability analyses still rely on labels or concept definitions, so the practical meaning of unsupervised interpretability may be more limited than the paper sometimes suggests.

**Strengths And Weaknesses:**

Strengths:

* This paper introduces a new concept, i.e., representer landmarks, and proposes KREPES, which introduces a second-order optimization framework based on Nyström approximation and GGN preconditioning.

* This paper introduces two novel metrics, i.e., Sample-Specific Influence Score and Conceptual Influence Profile, to quantify the transparency of the learned representations.
* Experiments conducted on multiple datasets demonstrate the effectiveness of the proposed method

Weaknesses:

* The paper argues that the learned SSL representation relies on demographic proxies, supported by the agreement gap analysis. However, the main paper does not provide enough detail on how the agreement gap is defined, how robust the conclusion is, or whether statistical variation was examined. At present, the evidence more clearly supports a feature-importance observation than a fairness conclusion.
* While criticizing the black-box nature of end-to-end deep SSL models, KREPES actually abandons joint deep feature learning entirely and trains a linear shallow projection model based on a fixed kernel (optimizing only the coefficient matrix $\\tilde{A}$). Therefore, the interpretability seems to belong to the proposed shallow kernel model, not the original deep SSL models. This paper does not provide sufficient explanations or evidence to justify why building a transparent kernel surrogate can explain modern deep SSL models.
* There have been some works that utilize kernel methods and representation theorems for sample attribution or interpretation of SSL, e.g., [1-3] that have not been cited. What are the advantages of this paper compared to them.
[1] Harilal, N., Rege, A. K., Bafghi, R. A., Raissi, M., & Monteleoni, C. (2024). Where Did Your Model Learn That? Label-free Influence for Self-supervised Learning. arXiv preprint arXiv:2412.17170.
[2] Simon, J. B., Knutins, M., Ziyin, L., Geisz, D., Fetterman, A. J., & Albrecht, J. (2023, July). On the stepwise nature of self-supervised learning. In International Conference on Machine Learning (pp. 31852-31876). PMLR.
[3] Song, Y., Jang, S., Katabi, D., & Son, J. (2023). Unsupervised object localization with representer point selection. In Proceedings of the IEEE/CVF International Conference on Computer Vision (pp. 6534-6544).
* The authors state that the eNTK used for CIFAR-10 is based on a "ResNet34 with pre-trained on ImageNet weights". Since the kernel has already acquired highly expressive semantic features through supervised pretraining, the "semantic coverage" subsequently exhibited by the model may be largely attributable to these supervised weights rather than the unsupervised KREPES optimization.
* While Section 5 highlights the training efficiency of KREPES, the current comparison omits the upfront computational cost of generating the eNTK. The paper does not provide detailed explanations or discussions to justify why this upfront computational overhead can be omitted, or why this initial cost would be acceptable in practice. Also, the authors selected optimal SSL loss hyperparameters by training a linear classifier on a labeled validation set without experiments on unsupervised learning paradigm. The ablation studies regarding important $m$ and $h$ are also required.
* There are lots of typos:
* there are many spelling errors, e.g., "underperform" should be "underperforms", "at Scalable" should be "at Scale", "a a" should be "a"
* inconsistent capitalization, e.g., Appendix X and appendix X.
* miss lots of ".", e.g., in L99-100 with "optimization Our framework", end of Eq.8
* the formulas are sometimes numbered, and sometimes not.
* Eq.1 and Eq.6 are outside the boundary.

---

> ### Author Rebuttal · Authors · 2026-03-31
>
> # W1: Definition & Robustness of Agreement Gap
> We thank the reviewer for their suggestion. Our method focuses on the feature-importance observation. We provide the following and will add in the revision:
>
> 1. **Definition.**
>  We introduce the Agreement Gap to quantify feature importance by measuring alignment between a test sample and its influential landmarks vs. random baselines. For feature $f$, similarity is
> $$v_f(x_t, z) = 1 - \min\left(\frac{|x_{t,f} - z_f|}{\Delta_f}, 1\right), \quad \Delta_f = \max(X_f)-\min(X_f).$$
> The Agreement Gap is
> $$\text{AG}f = \mathbb{E}{x_t}\Big[\frac{1}{K} \sum_{z \in \mathcal{L}{top}(x_t)} v_f(x_t,z) - \frac{1}{K} \sum_{z \in \mathcal{L}_{rand}} v_f(x_t,z)\Big].$$
> A larger $\text{AG}_f$ indicates stronger emphasis of feature $f$ in the SSL latent space.
>
> 2. **Full-Distribution Evaluation & 95% CI.**
> We recomputed the Agreement Gap on full test sets (Adult: 200K; Covertype: 116,202; K=5), reporting 95% CIs in Fig. 4. The tight error bars confirm that observed effects (e.g., gender/relationship dominance) reflect true SSL geometry rather than sampling noise.
>
> # W2 & Limitation: Validity of Kernels/NTKs as NN Approximators
> This critique overlooks the established foundation of eNTK surrogate modeling and the Representer Theorem linking NNs and kernels [6,7] (see response to W1, reviewer PcWw). Our goal is quantitative interpretability via kernel-based surrogates, a standard approach in supervised learning using representer/influence methods [1,2,3,8]. Further faithfulness results are provided in our response to W1 & Key Q1 (reviewer Dsic).
>
> # W3: Comparison with the Referred Works
> KREPES provides advantages in scalability, generality, and test-point interpretability.
> - Harilal et al., 2024:
>     1. _Test-Point vs. Self-Influence:_  Their method measures self-influence $\mathcal{I}(x_{train}, x_{train})$ and cannot explain unseen test points. KREPES performs test-point attribution $\mathcal{I}(x_{train}, x_{test})$, explaining how landmarks shape new representations.
>     2. _Scalability:_ They rely on explicit $H^{-1}$ computation, which is expensive/unstable. KREPES uses Nyström + GGN preconditioning, scaling to >1.2M samples (ImageNet).
>     3. _Causal Influence:_ Standard influence $\nabla_\theta f(z_{test})^\top \left( -H_{\theta}^{-1} \nabla_\theta L_{SSL}(z_{train}) \right)$ requires $H^{-1}$. In KREPES, term1: The gradient of the test representation with respect to the $l$-th row of $A$ is our local similarity: $k(x_l, x_t)$.
> Term 2: We train $A$ using GGN updates: $\Delta A = -H_{GN}^{-1} \nabla_A L$. Hence,
>         $k(x_l, x_t) \cdot A_{l,:}$
>         Taking $L_2$ norm of this influence vector yields our Eq. 5.
> - (Simon et al., 2023): This is a theoretical work on training dynamics of SSL and does not provide interpretability of learned representations.
> - (Song et al., 2023): Leverages representer points for unsupervised object localization, whereas KREPES provides interpretability of SSL models. This work can be a downstream application of our method.
>
> # W4: eNTK of a pretrained ResNet on ImageNet
> 1. All tabular datasets (Adult, Covertype, Higgs, Bank) use eNTK from randomly initialized models (Appendix C).
> 2. The framework is not restricted to pretrained ResNets; any model’s eNTK can be used. Our additional Solo-learn CIFAR-10 (Barlow Twins) result further supports this (W3 & Key Q2 response).
> 3. The reviewer's comment is not clear to us. The goal is not to claim kernel superiority over NNs, but to enable systematic interpretability of SSL; KREPES uses SSL optimization solely to identify influential landmarks.
>
> # W5: eNTK Calculation vs. Training Cost of KREPES
> 1. *Figure 5 Clarification:* the kernel is computed on-the-fly during training, so no separate upfront cost; the figure evaluates scalability of our Nyström-accelerated solver vs. full kernel computation.
> 2. Our focus is not kernel vs. NN performance, but an interpretable approximation; the added eNTK cost is the price of interpretability and is kept practical via Nyström acceleration.
> 3. A one-time cost is acceptable: (i) Nyström reduces $O(n^2)$ to $O(nm)$ ($m \ll n$), and (ii) optimized parallelization (Appendices A, L) yields ~2h (ResNet-34), ~1h (ResNet-18), far below prior NTK costs (1000 A100 GPU hours [10]) on Cifar-10.
> 4. *Hyperparameter Tuning*: follows standard SSL practice (SimCLR, BYOL, VICReg); both KREPES and baselines use identical tuning, ensuring fair comparison of accuracy gap ($\Delta$) and fidelity.
> # W6: Typographical Errors
> We thank the reviewer for pointing this out. We will carefully fix these in the revision.
>
> # Limitations
> - Addressed in W2 & Limitations.
> - The reviewer appears to misunderstand our method: it does not depend on labels or concept definitions. The influence-function pipeline is fully self-supervised; labels/concepts are used only for post hoc evaluation of representer landmark quality.
>
> # References: Please see response to reviewer 2e3x

---

> > ### Author Rebuttal · Reviewer_LQUW · 2026-04-03
> >
> > **Response to A1:**
> >
> > Thank you for the clarification. While the rebuttal now defines Agreement Gap more clearly and shows that the measurement is statistically stable, it still mainly supports a feature-importance observation rather than a fairness claim.
> >
> > A larger Agreement Gap indicates that a feature is more strongly reflected in the SSL representation geometry, but this does not by itself demonstrate unfairness, harmful reliance, or disparate downstream effects. The added confidence intervals only show that the phenomenon is stable, not that it has fairness implications. As such, the current evidence remains descriptive of representation structure, and the fairness conclusion still appears overstated relative to what is actually shown.
> >
> > **Response to A2:**
> >
> > Thank you for the clarification. I remain unconvinced. The response largely appeals to the existence of kernel-surrogate literature, but this does not answer the actual concern: why does a shallow fixed-kernel model explain the original deep SSL model, rather than simply replacing it with a more interpretable proxy.
> >
> > The theoretical references cited do not by themselves establish faithfulness in this setting, especially since modern SSL relies on jointly learned deep representations rather than fixed-kernel shallow inference. As written, the interpretability still appears to belong to the surrogate model, not to the original deep SSL model.
> >
> > **Response to A3:**
> >
> > Can you provide some empirical demonstrations？
> >
> > **Response to A5:**
> >
> > The rebuttal provides useful implementation detail, but it does not yet fully resolve the accounting issue around kernel computation. In particular, “on-the-fly” computation does not imply zero overhead, and the current response does not clearly state whether the reported efficiency numbers include all kernel-related steps. In addition, the labeled-validation protocol remains difficult to reconcile with a strictly unsupervised framing, even if it is commonly used in practice. Finally, the requested ablation analysis on key hyperparameters is still missing.

---

> > > ### Author Response · Authors · 2026-04-07
> > >
> > > # A1: Replacing fairness with feature importance
> > >  We agree with the reviewer’s assessment. Since our work's primary focus is interpretability, we will remove all fairness claims in the revision and reframe as "feature importance observations". We will also include a detailed explanation of feature agreement.
> > > # A2: Reviewer fundamentally disagrees with the faithfulness of kernel approximations
> > > We clarify that [6, 7] were cited to establish the theoretical neural network-kernel link, whereas [1, 2, 3, 8] explicitly validate NTKs for interpretability. Specifically, [1] establishes faithfulness using same exact methodology as ours.
> > >
> > > Despite the additional faithfulness results, the reviewer disagrees with the faithfulness of kernel approximations; hence, further response is unlikely to be satisfactory. We can only note: this perspective has been taken in ICML, ICLR, NeurIPS papers [1, 2, 3, 8], albeit in supervised setting, and extending it to SSL is the core contribution of ours.
> > > # A3. Empirical Demonstrations
> > > We provide empirical evidence in the following (anonymized single-page, figure-only) PDF containing the plots referenced in this response: https://figshare.com/s/5756a99417a9acbf86d4
> > >
> > > **1. Representer Landmarks in Latent Space (Plot a)**
> > >  Fig. 3 in [11]: Plot (a) illustrates our metric’s behavior in representation space. KREPES reveals a density-dependent dual behavior in SSL:
> > > For highly separable classes, highest influential images act as prototypes (cluster centers), anchoring the global geometry. For entangled classes, it mimics self-influence: points near overlapping boundaries exhibit high geometric tension and large gradients, assigning high influence to boundary-defining samples.
> > >
> > > **2. Semantic Separation via Test-Time Attribution (Plots b & c)**
> > > Beyond the scope of [11], KREPES enables direct attribution for unseen samples. Plots (b)-(c) show random test samples with positive/negative influence edges. Negative edges (red) often link visually similar but different-class samples (e.g., flying bird in sky background driving a "bird" feature, must be suppressed to embed an "airplane"), reflecting SSL objective's repulsive forces, and revealing how KREPES exposes model's dynamics for distinguishing confusable clusters.
> > >
> > > **3. Instance-Specific Attribution (Manuscript Figs. 2&7)**
> > > Figure 4 in [11]: The key distinction in Figs. 2&7 is that our framework performs localized test attribution, identifying landmarks responsible for a given test embedding, rather than general training self-influence.
> > >
> > > **4. Highest vs. Lowest Influential Samples (Plot d)**
> > > Figs. 4& 5 in [11]: Plot (d) contrasts the most and least influential samples: top ones are clear, prototypical instances with distinct subjects, while the lowest are noisy and redundant, confirming that KREPES filters noise and anchors the SSL manifold on semantic prototypes.
> > >
> > > **5. Data Removal Ablation**
> > > Fig. 6 in [11]: we evaluated downstream accuracy after removing top, random, and lowest $k$ influential samples from training set.
> > >  ### Table 1:Downstream Accuracy. CIFAR-10 BT, after removing $k$ examples based on influence scores.
> > > |($k$)|No Top-$k$|No Random-$k$|No Lowest-$k$|
> > >  | :--- | :---: | :---: | :---: |
> > >  | **2,000** | 84.22 | 81.40| 74.33|
> > >  | **8,000** | 81.47| 80.27| 73.73|
> > >  | **16,000** | 76.27| 70.65| 69.05|
> > >
> > > This reveals a counterintuitive SSL effect: removing low-scoring points harms accuracy far more than removing top examples. Unlike supervised models that depend on clear prototypes, SSL relies on a dense mass of hard examples to shape the contrastive space. Further, unlike [11], once removals exceed the optimal landmark count ($m < 2000$ for CIFAR-10), the gap between removing top vs. low points shrinks, showing that KREPES isolates a sparse set of influential landmarks, while the remaining data is largely redundant.
> > >
> > > **6. Feature Importance & Auditing (Manuscript Fig. 4)**
> > > Fig. 8 in [11]: While analysis in [11] are restricted to training representation space, our Agreement Gap analysis (Fig. 4) conducts auditing at inference, exposing SSL model's implicit reliance on demographic proxies (Adult) as an auditing tool for unseen data.
> > > # A5: Ablation on $m, h$ CIFAR-10, BT | Kernel computation
> > > - $h=512$ fixed
> > > |($m$)|Accuracy(%)|
> > > |:---:|:---:|
> > > |100|33.41|
> > > |200|80.3|
> > > |500|82.77|
> > > |1000|90.62|
> > > |2000|87.84|
> > >
> > > - $m=1000$. fixed
> > > |($h$)|Accuracy(%)|
> > > |:---:|:---:|
> > > |128|89.32|
> > > |256|89.64|
> > > |400|89.71|
> > > |512|90.62|
> > > |1024|90.56|
> > >
> > > -  We used an on-the-fly setup in Fig. 5 to ensure a fair comparison with the full-kernel baseline; the reported times include all kernel-related steps, confirming the scalability of our Nyström method over the full kernel approach. The previously reported times for CIFAR-10 likewise cover the full SSL pipeline, including kernels of augmented samples.
> > >
> > > [11] Harilal, N., Rege, A. K., Akbarian Bafghi, R., Raissi, M., & Monteleoni, C. (2024). Where did your model learn that? Label-free influence for self-supervised learning. arXiv.

---

### Official Review · Reviewer_Dsic · 2026-03-10

**Soundness:** 4
**Presentation:** 3
**Significance:** 3
**Originality:** 3
**Overall Recommendation:** 5
**Confidence:** 3

**Summary:**

This paper introduces KREPES, a framework for intrinsically interpretable self-supervised representation learning. The main idea is to approximate an SSL backbone with an empirical NTK, express learned representations through representer landmarks via a kernel formulation, and optimize the resulting model at scale using Nyström approximation, principled initialization, landmark selection, and preconditioned optimization. The paper also proposes two interpretability primitives, Sample-Specific Influence Score and Conceptual Influence Profile, and evaluates them on image and tabular datasets, including large-scale settings up to 1M+ samples.

**Compliance With Llm Reviewing Policy:**

Affirmed.

**Final Justification:**

The authors addressed most of my concerns, but the work still falls short of true interpretability. However, considering that this work represents a step forward in interpretability within self-supervised learning and is supported by solid experimental evidence, I have decided to raise my rating.

**Key Questions For Authors:**

1. Can the authors provide a more direct faithfulness evaluation between KREPES explanations and the original SSL network, beyond downstream accuracy gap? For example, is there agreement in top influential training samples, or similar behavior under removal / perturbation of top-ranked landmarks?

2. The paper positions KREPES as intrinsic interpretability, but the Conceptual Influence Profile requires external concept sets and CAVs. Would the authors consider reframing this component as a post-hoc concept probe built on top of an intrinsic sample-level explanation framework?

3. BYOL appears to be the least well matched to the current approximation/optimization recipe. Do the authors believe this is mainly a proxy issue, a preconditioning issue, or something specific to asymmetric predictor dynamics?

4. Since labels are used for validation during hyperparameter tuning, did the authors try any label-free model selection criteria, such as spectral diagnostics, representation-uniformity measures, or landmark consistency statistics?

**Limitations:**

yes

**Strengths And Weaknesses:**

**Strengths**

1. The paper addresses a real gap in the literature: existing SSL explanation approaches are often post-hoc or domain-specific, whereas this work aims to make the learned representation itself interpretable.

2. The paper combines several ingredients into one reasonably unified pipeline: eNTK approximation of SSL backbones, representer-based parameterization of latent space, sample-level and concept-level interpretability metrics, Nyström approximation, principal-component initialization, and second-order / preconditioned optimization for non-convex SSL losses. This level of integration is a clear strength.

3. The paper does not only show cherry-picked nearest neighbors. It introduces quantitative landmark consistency analysis, where influential landmarks align well with semantics; for example, on Adult, Precision@1 reaches 0.872 and Majority@10 reaches 0.921. These numbers make the interpretability claims more credible than purely qualitative visualizations.

4. The feature-importance / fairness-auditing experiment on Adult is compelling: the model appears to rely strongly on sensitive demographic proxies such as gender and relationship, rather than only on merit-related factors. This is exactly the kind of failure mode that intrinsic interpretability should help surface.


**Weaknesses**

1. The method explains an eNTK surrogate, not the original neural SSL model directly. The paper does evaluate proxy quality using downstream accuracy gaps, but the results are mixed, especially for BYOL, where the gap is consistently negative in several cases. This means the interpretability claim is strongest for the surrogate and only indirectly supported for the original model.

2. The decomposition into global importance and local similarity is reasonable, and the concept profile is elegant, but the paper does not yet provide strong theoretical analysis. The metrics are convincing as tools, but less convincing as fully characterized interpretability objects.

3. The Conceptual Influence Profile depends on manually defined positive/negative concept sets and CAV computation. That is a fair and useful evaluation device, but it is more accurate to describe it as a concept-level analysis layered on top of the intrinsic representer framework, rather than a purely self-contained form of interpretability.

4. The appendix states that downstream evaluation uses only 10% of labeled data for the linear classifier rather than standard full linear probing, and the setup is explicitly framed as a controlled comparison between KREPES and the corresponding NN surrogate. This is fine for validating the proxy, but it weakens claims about absolute competitiveness in standard SSL settings.

---

> ### Author Rebuttal · Authors · 2026-03-31
>
> # W1 and Key Q1: More direct Faithfulness Evaluation, BYOL  Performance Drop
> Using kernel surrogates to approximate and explain deep networks is well established in supervised learning [1,2,8]; our main contribution is extending this to SSL. It is also known that interpretable approximations incur an inherent trade-off [4,5]. This simplification is necessary, as computing influence scores directly for deep models is intractable due to the $H^{-1}$ term (see discussion in W3,  reviewer LQUW).
>
> To further demostrate our interpretability transfers to original SSL model, we conducted two faithfulness evaluations:
> 1. Downstream Decision Fidelity (Kendall-$\tau$): Following Engel et al. (2024), we measure rank correlation between original SSL predictions and the KREPES surrogate, showing strong agreement (CIFAR-10: 0.87, Adult: 0.84, Covertype: 0.77).
> 2. Inference-Time Latent Ablation
> Inference-Time Latent Ablation: Using a 50-NN classifier on frozen embeddings, we remove the top-10 KREPES landmarks and compare confidence drops against random deletions over the full test set.
> | Dataset (N) | Base Conf | Drop (Rand / KREPES) |
> |---|---|---|
> | Adult (200k) | .7542 | .0002 / **.0453** |
> | CIFAR-10 (10k)| .8126 | .0011 / **.0651** |
>
> While the model is robust to random deletions, removing KREPES-identified landmarks causes a sharp confidence drop, indicating that KREPES captures the key support vectors driving the model’s predictions.
> # W2: Theoretical characterization of the interpretability metrics (Eq. 5 and Eq. 6).
> Our interpretability metrics are not heuristic. The Sample-Specific Influence Score (Eq. 5) and Conceptual Influence Profile (Eq. 6) arise directly from the stationary conditions of regularized empirical risk minimization in RKHS.
>
> 1. Eq. 5: Standard influence $\mathcal{Inf}{(z_l \rightarrow z_t)} = \nabla_\theta f(z_t)^\top \left( -H_{\theta}^{-1} \nabla_\theta L_{SSL}(z_l) \right)$ requires $H^{-1}$. In KREPES, term1: The gradient of the test representation with respect to the $l$-th row of $A$ is our local similarity: $k(x_l, x_t)$.
> Term 2: We train $A$ using GGN updates: $\Delta A = -H_{GN}^{-1} \nabla_A L$. Hence, $k(x_l, x_t) \cdot A_{l,:}$. Taking $L_2$ norm of this influence vector yields our Eq. 5.
>
>  At the same time, Eq. 5 is the RKHS/self-supervised analogue of generalized representer theorems defined in supervised setting [6,7] as follows:
> - Global importance ($\||A_{l,:}\||_2$): from KKT conditions, captures SSL loss sensitivity to landmark $x_l$, independent of $x_t$.
> - Local importance ($\||k(x_l, x_t)\||$): RKHS similarity, analogous to localized inner products [2].
> 2. Eq. 6: Let $v_c \in \mathbb{R}^h$ be a Concept Activation Vector. The concept activation is: $  \langle f(x_t), v_c \rangle = \left\langle \sum_{l=1}^{m} A_{l,:} k(x_l, x_t), v_c \right\rangle,$
>     by the linearity: $$ \langle f(x_t), v_c \rangle = \sum_{l=1}^{m} k(x_l, x_t) \langle A_{l,:}, v_c \rangle = \sum_{l=1}^{m} Score(l \rightarrow t, c) $$ The $Score(l \rightarrow t, c)$ is not a heuristic. It is the additive decomposition of the total semantic concept activation into the per-landmark directional force in the representation space.
>
> # W3 and Key Q2: Clarification on Intrinsic vs. Concept-Based Interpretability
> We agree that CAVs rely on external, human-defined concepts, hence labeling the Conceptual Influence Profile (CIP) as purely "intrinsic" blurs is imprecise. We will revise Section 3.2 accordingly.
>
> # W4: Downstream on 100% train data
> We ran full downstream evaluation on CIFAR-10; using all labels improves accuracy by ~0.5% (Barlow Twins: 91.18%, VicReg: 90.35%, SimCLR: 90.13%). We will extend this to other datasets and losses in the revision.
>
> # Key Q3: BYOL Performance Drop Clarification
> We attribute the drop to BYOL’s asymmetric, non-stationary objective. For symmetric losses (e.g., Barlow Twins), the loss is NLLS,
> $\mathcal{L}(\theta)=\tfrac{1}{2}\||r(\theta)\||^2,
> H \approx J^\top J,$
> so the GGN approximation is accurate and the preconditioner captures curvature well. In contrast, BYOL $L = \|| q_\theta(z_\theta) - z_\xi \||^2,$ violates these assumptions due to a moving target, predictor head, and stop-gradient, yielding a non-symmetric, time-varying curvature that GGN cannot capture, leading to suboptimal alignment.
>
> # Key Q4:
> We agree a zero-label pipeline is ideal. Here, we used 10% linear validation for hyperparameter tuning following standard SSL (e.g., SimCLR, BYOL) to ensure fair comparison with baselines. However, our framework naturally supports label-free alternatives. In fact, we already perform spectral analysis via the eigenvalue spectrum of $(\tilde{A}^\top \tilde{A})$ (Section 5, Fig. 6) to assess collapse. This, together with representation uniformity and our Landmark Class Coverage metric ($\kappa$), could enable fully unsupervised model selection (e.g., via effective rank maximization). We will add this as future work.
>
> # References: Please see response to reviewer 2e3x

---

> > ### Author Rebuttal · Reviewer_Dsic · 2026-04-03
> >
> > Thanks to the authors' rebuttal, some of my concerns have been resolved, so I've decided to keep the current positive score for now. As for whether to raise the score, I need to consider it further. The authors could continue to refine their responses to weaknesses and questions below the acknowledgment, and more directly and specifically explain the value and significance of their work, as well as its advantages compared to other methods.

---

> > > ### Author Response · Authors · 2026-04-07
> > >
> > > We thank the reviewer for the continued engagement. To resolve their remaining inquiries, we articulate KREPES's advantages, demonstrate its utility for label-free tuning, and provide finalized 100% data evaluations. We also provide additional empirical evidence in the following (anonymized single-page, figure-only) PDF containing the plots referenced in this response: https://figshare.com/s/5756a99417a9acbf86d4
> > >
> > > # 1. Full Downstream Evaluation & Solo-Learn Performance (W4)
> > > We evaluated the linear probe using 100% labeled training data across datasets and losses.
> > > ### Table 1: Downstream Accuracy across losses and datasets. (10% Labels / 100% Labels)
> > > | Dataset | Barlow-Twins | VICReg | SimCLR | BYOL |
> > > | :--- | :--- | :--- | :--- | :--- |
> > > | CIFAR-10 | 90.31 / 90.62 | 89.88 / 90.35 | 89.54 / 90.13 | 88.76 / 89.03 |
> > > | Adult | 83.78 / 84.08 | 83.60 / 83.82 | 83.67 / 83.90 | 83.36 / 83.58 |
> > > | CoverType | 71.32 / 71.45 | 72.85 / 73.03 | 75.17 / 75.34 | 70.78 / 70.97 |
> > > | Higgs | 67.66 / 67.75 | 68.70 / 68.74 | 69.50 / 69.59 | 68.24 / 68.31 |
> > >
> > > To demonstrate KREPES scales with heavily optimized backbones, we evaluate it with standard solo-learn pre-trained parameters on CIFAR-10:
> > > ### Table 2: Downstream Accuracy across losses on CIFAR-10 (100% Labels)
> > > | Loss |Ours| Solo-learn|
> > > |-|-|-|
> > > |BT| 90.62| 91.18|
> > > |VICReg| 90.35| 90.97|
> > > |SimCLR| 90.13| 90.36|
> > > |BYOL| 89.03| 90.56|
> > >
> > > These confirm that the performance ceiling of our interpretable surrogate is governed by underlying deep SSL backbone, effectively closing the previously observed performance gaps.
> > > # 2. Label-Free Hyperparameter Tuning (Key Q4)
> > > To address the zero-label pipeline, we refer to **Plots (e) and (f)**. Tracking the spectral entropy of the KREPES matrix $\tilde{A}$ across a logarithmic grid of the Barlow Twins regularization parameter ($\lambda$), we find that that its peak aligns with optimal linear probe accuracy. This empirically proves KREPES extends beyond post-hoc attribution and can serve as a label-free diagnostic for SSL model selection.
> > >
> > > # 3. Value, Significance, and Advantages over Baselines
> > > The core contribution of KREPES is to transform deep SSL models into tractable proxies that enable instantaneous test-time attribution. A related approach has been proposed in the supervised setting [1]; however, to the best of our knowledge, no influence-score-based, systematic interpretability method for test-time attribution currently exists in SSL.
> > > Further, to see our advantages over [11]  on self-influence score, we direct the reviewer to the attached PDF, and our corresponding response (A3) to Reviewer LQUW:
> > > - **Test-Time Attribution vs. Self-Influence:** Hariral et. al. [11] rely on computationally massive inverse-Hessian, as well as restricted to computing self-influence within the training set. Since KREPES amortizes the GGN optimization into $\tilde{A}$, it computes causal influence for unseen test points at inference time via a forward pass.
> > > - **Optimization Influence vs. Spatial Proximity:**
> > > Since exact test-time attribution is intractable for methods like [11], one might default to clustering metrics (e.g., $k$-NN), assuming nearest neighbors are most explanatory. However, Plot (b) shows this is insufficient in SSL. For a test query (e.g., the orange star corresponding to the automobile shown in Plot c), KREPES identifies influential landmarks that originate from further apart across the manifold, rather than within its immediate local cluster, correctly isolating the structural anchors that drove the optimization of the SSL loss (as the influence score originates from the gradient and $H^{-1}$), ignoring geometrically close but optimization-redundant samples.
> > > - **Quantifying Negative Influence:**
> > > While [11] focuses solely on positive self-influence, KREPES also captures negative influence, identifying training landmarks that exerted repelling gradient forces on the test representation. Distance-based clustering metrics are also strictly non-negative and are incapable of surfacing these oppositional forces.
> > > - **Label-Free Hyperparameter Tuning via Spectral Diagnostics:**
> > > Beyond post-hoc attribution, KREPES encodes the structure of the SSL representation space. Plots (e) and (f) track the spectral entropy of ($\tilde{A}$) across a logarithmic grid of the Barlow Twins regularization parameter ($\lambda$). The peak of this unsupervised spectral entropy aligns with the optimal linear probe accuracy. This demonstrates that KREPES can be utilized as a label-free diagnostic for SSL model selection, addressing a major practical limitation in unsupervised learning pipelines.
> > >
> > > We will integrate these findings and visual comparisons into the revised manuscript.
> > >
> > > [11] Harilal, N., Rege, A. K., Akbarian Bafghi, R., Raissi, M., & Monteleoni, C. (2024). Where did your model learn that? Label-free influence for self-supervised learning. arXiv.

---

### Official Review · Reviewer_2e3x · 2026-03-13

**Soundness:** 3
**Presentation:** 3
**Significance:** 4
**Originality:** 3
**Overall Recommendation:** 5
**Confidence:** 1

**Summary:**

This paper proposed an interpretability method for self-supervised pre-trained models. Because self-supervised learning (SSL) does not rely on ground-truth labels, the method adopts a sample-based interpretability approach, where a test sample is explained through its relationships to a few training samples.

The core idea is to represent a sample of interest as a weighted combination of similarities to a subset of training samples, computed using an empirical neural tangent kernel (eNTK). The training process directly optimizes the representer coefficients as model parameters, using an SSL objective (e.g., SimCLR).

The formulation builds on an adaptation of the representer theorem to the SSL setting. Within this framework, the authors introduce several interpretability metrics. These include class coverage, defined as the minimum number of top-ranked representers required to span all ground-truth classes; a sample-specific influence score, which decomposes a representer’s influence on a test sample into global and local importance components; and a conceptual influence score, which measures how a representer’s contribution projects onto a predefined concept vector.

Since directly computing the kernel representation is computationally prohibitive for the large datasets typical in SSL, the method incorporates several efficiency improvements. In particular, it uses a Nyström-based approximation combined with random Fourier features to approximate the kernel, along with an improved initialization based on principal components and optimization via an approximate second-order method.

The authors empirically validate the approach on multiple SSL objectives, namely SimCLR, BarlowTwins, VICReg, and BYOL.

**Compliance With Llm Reviewing Policy:**

Affirmed.

**Final Justification:**

I have read the reviews and rebuttal to all reviewers.

While I agree with the limitation pointed out by *Reviewer LQUW* on whether a fixed-kernel model is a good surrogate for explaining the original deep SSL model, this paper provides a reasonable first step towards the interpretability of SSL models, and thus still makes a valuable contribution in the reviewer's opinion. I thus keep my recommendation to accept.

**Key Questions For Authors:**

- Could the authors provide a simplified or more conceptual version of Figure 1 to make the method easier to understand at a glance?

**Limitations:**

yes

**Strengths And Weaknesses:**

### Strengths
1. Interpretability of SSL methods is a relevant problem, as those SSL methods are increasingly used in a wide range of downstream applications.
2. The approach is, to the best of the reviewer's knowledge, the first to provide a scalable interpretability method for self-supervised learning.

### Weaknesses
1. The overview illustration (Figure 1) is quite dense and includes many equations, making it difficult to quickly grasp the main idea of the method. A more illustrative figure could help readers better understand the core ideas.

Due to limited knowledge of this research subfield, the reviewer could not assess the soundness of the method and of the empirical results.

---

> ### Author Rebuttal · Authors · 2026-03-31
>
> # Global References: Used Throughout the Author Responses
>
> [1]Engel, A., Wang, Z., Frank, N., Dumitriu, I., Choudhury, S., Sarwate, A. D., and Chiang, T. Faithful and efficient explanations for neural networks via neural tangent kernel surrogate models. International Conference on Learning Representations, ICLR 2024.
>
> [2]Tsai, C.-P., Zhang, J., Yu, H.-F., Chien, E., Hsieh, C.-J., and Ravikumar, P. K. Representer point selection for explaining regularized high-dimensional models. In International Conference on Machine Learning, ICML, 2023.
>
> [3]Yeh, C.-K., Kim, J., Yen, I. E.-H., and Ravikumar, P. K. Representer point selection for explaining deep neural networks. In Bengio, S., Wallach, H., Larochelle, H., Grauman, K., Cesa-Bianchi, N., and Garnett, R. (eds.), Advances in Neural Information Processing Systems, 2018.
>
> [4]Laber, E. S. and Murtinho, L. On the price of explainability for some clustering problems. In Proceedings of the 38th International Conference on Machine Learning (ICML), 2021.
>
> [5]Moshkovitz, M., Dasgupta, S., Rashtchian, C., and Frost, N. Explainable k-Means and k-Medians Clustering. In Proceedings of the 37th International Conference on Machine Learning (ICML).
>
> [6]Bohn, B., Rieger, C., Griebel, M. A Representer Theorem for Deep Kernel Learning. Journal of Machine Learning Research (JMLR), 2019.
>
> [7] Unser, M. A Representer Theorem for Deep Neural Networks. Journal of Machine Learning Research (JMLR), 2019.
>
> [8]Park, S.M., Georgiev, K., Ilyas, A., Leclerc, G. &amp; Madry, A. TRAK: Attributing Model Behavior at Scale. Proceedings of the 40th International Conference on Machine Learning (ICML), 2023.
>
> [9]Koh, P. W. and Liang, P. Understanding black-box predictions via influence functions. In International Conference on Machine Learning (ICML), 2017.
>
> [10]Shankar, V., Fang, A., Guo, W., et al. Neural Kernels Without Tangents. In International Conference on Machine Learning (ICML), 2020.
>
> # W1 & Key Q1: Clarity of Firgure1
> We thank the reviewer for their positive assessment.
> We are currently trying to find a better solution for a clearer illustration of our framework. We will update this in the revision.

---

> > ### Author Rebuttal · Reviewer_2e3x · 2026-04-04
> >
> > Thank you for considering improving the illustration.

---

### Official Review · Reviewer_PcWw · 2026-03-15

**Soundness:** 2
**Presentation:** 2
**Significance:** 3
**Originality:** 3
**Overall Recommendation:** 3
**Confidence:** 4

**Summary:**

The paper introduces explanatory methods for self supervised strategies. The authors introduce a method called KREPES that learns interpretable representations regardless of the SSL objective. They do this by identifying a concept called Representer Landmarks that are representations of unlabeled training samples that are considered influential. These Landmarks can serve as a proxy for an explanation. They do this through two novel metrics called the Sample Specific Influence Score and Conceptual Influence Profile.

**Compliance With Llm Reviewing Policy:**

Affirmed.

**Final Justification:**

The rebuttal addressed my main concerns. I thank the authors for the detailed and thoughtful rebuttal. Both the tables addressed my concern regarding more standardized evaluation setups. I understand and accept the tradeoff between SOTA and interpretable models. The data curation and pruning table is certainly a more direct evaluation regarding feature importance (I would not use the word causal support vectors since it requires insertion/deletion experiments and this seems only deletion). As such I am increasing my score.

I agree about the point of instance-level vs dataset-level techniques. However, the definition of interpretability, as defined by the authors is quite narrow since clustering can directly be used to define instance-level distances. I have gone through the other reviews as well and believe that some of the claims in the paper can be tempered.

**Key Questions For Authors:**

Please see the Weaknesses. Additionally,

1. How are the concepts used in your metric generated? I thought such an idea wouldn’t be accessible in SSL.

2. Is it possible to try using the parameters in the solo-learn library to verify your method?

**Limitations:**

The extensive ablations in the appendix can be qualitatively discussed better within the conclusion section.

**Strengths And Weaknesses:**

# Strengths

The authors address an important problem of interpretability in Self Supervised Learning. Since SSL does not include labels, it is difficult to understand what the model is learning. Furthermore, traditional explanatory techniques rely on predictions as a mechanism to generate explanations. They address this problem through the idea of representative samples that one can trace back to the training set as a definition of interpretability. This is motivated by prior theoretical work on kernels. It also introduces an interesting method that decomposes SSL methods into the format of kernel learning.

# Weaknesses

1. The proposed method and the corresponding analysis does not function as sufficient validation that the authors produced a viable explainability strategy for SSL. The paper is motivated by the idea that we can produce more interpretable SSL representations without a significant loss in SSL performance or perhaps better performance in specific cases. In general, it is difficult even within the supervised setting to claim that a representation is more interpretable. In this case, by using a kernel strategy with parameters we are able to identify samples that are considered landmarks. The authors then present a series of figures and tables that show that these landmarks correspond well with relevant semantic concepts. In this case, this is stating that the most important samples corresponding to relevant classes is considered a more interpretable space. An issue with validation approach is that standard self supervised learning alone can already be implicitly viewed as producing this idea of landmarks. A variety of works have shown that self supervised learning produces representations that naturally cluster into semantic concepts irrespective of labels. This includes [1][2] that study the clustering properties of SSL models. In both of these works, simple clustering analyses indicate that SSL representations are interpretable in the sense that they are organized in semantic regions. It is not surprising to me that it is possible to identify samples that are reliably in a targeted class with SSL representations. A more convincing strategy would be to compare the technique's Hit rate or correlation with a metric that is naturally measured in representation learning papers such as cosine similarity or neighborhood consistency [3]. I think the way it is discussed now makes the claim of a more interpretable representation space hard to verify when compared with simply SSL working as intended.

2. One could also argue that kernel methods are naturally more interpretable. However, this is treated as a fact in the paper and not treated as something that needs to be demonstrated. Ideally, this interpretability should be leveraged in some manner to improve some aspect of the overall SSL process. However, the paper does not do a good job of demonstrating the significance of this interpretability in the context of SSL.

3. I believe that the empirical performance of the method can use further verification. The premise of the paper is that more interpretability is good as long as it doesn’t come with the trade-off of worse performance. The authors show tables where their method performs better or similarly to a baseline SSL strategy. However, I do not believe their tables are empirically rigorous enough to make these claims. For example, Table 3 shows comparisons with improvements on Barlow Twins and SimCLR with a performance score of 89.11 for baseline barlow twins and their method performing at 90.31 accuracy. However, SSL has standard benchmarks that the authors could have used to validate their method against instead of relying on these scores that are from difficult to verify sources. For example, in the solo-learn benchmark [4] Barlow Twins on Cifar-10 is reported as 92.10. This makes it difficult to verify whether their method is comparable to baseline SSL methods or their improvement is a function of their specific hyperparameters used for their comparison. This concern only increases further when looking at Table 4 where there are significant performance decreases by using their method. For example, BYOL with the kernel method decreases by 2 % on ImageNet. In my opinion, this is a significant decrease in performance for this dataset and is not negligible.

[1] Ben-Shaul, I., Shwartz-Ziv, R., Galanti, T., Dekel, S., & LeCun, Y. (2023). Reverse engineering self-supervised learning. Advances in Neural Information Processing Systems, 36, 58324-58345.

[2] Kokilepersaud, K., Kim, S., Prabhushankar, M., & AlRegib, G. (2025, February). Hex: Hierarchical emergence exploitation in self-supervised algorithms. In 2025 IEEE/CVF Winter Conference on Applications of Computer Vision (WACV) (pp. 1111-1121). IEEE.

[3] Park, Young-Jin, et al. "Quantifying representation reliability in self-supervised learning models." arXiv preprint arXiv:2306.00206 (2023).

[4] Da Costa, V. G. T., Fini, E., Nabi, M., Sebe, N., & Ricci, E. (2022). solo-learn: A library of self-supervised methods for visual representation learning. Journal of Machine Learning Research, 23(56), 1-6.

# Minor Weakness

The paper clarity can be improved. The paper touche across multiple fields and not a sufficient amount of motivation for each for me to follow. For example, it was unclear to me where the idea of a concept was coming from in the SSL context (i.e. without labels). Yet, the proposed metric seemed to rely on an alignment with a concept that should not exist in SSL analysis.

---

> ### Author Rebuttal · Authors · 2026-03-31
>
> # W1: Reviewer questions KREPES as a viable interpretability strategy
>
> - The reviewer questions whether kernel-based approximations (influence functions/representer points) provide meaningful interpretability in SSL, given that SSL representations already show semantic structure (e.g., clustering). However, clustering is only a qualitative, dataset-level analysis and does not provide a systematic, instance-level explanation of why a specific test point is represented in a certain way, nor does it generalize without downstream labels. In contrast, kernel-based methods offer a principled framework for post-hoc, instance-level explanations in supervised learning. To our knowledge, no such framework exists for SSL, and extending it is non-trivial (e.g., beyond regression-based settings and handling SSL objectives). Our work addresses this gap. Our goal is to address the lack of a systematic, quantitative, and general interpretability framework for SSL that explains individual predictions at inference.
>
> We disagree that clustering provides such interpretability: it only shows that representations are semantically organized, but does not attribute or audit individual decisions. For example, in our Adult dataset analysis, our influential landmarks enable quantifying gender bias at the representation level, which clustering cannot reveal.
>
>  - We thank the reviewer for the suggestion. Using SSL embeddings, we evaluated cosine similarity (CS) of nearest-neighbor consistency over the same KREPES landmarks on 200K Adult test samples, using identical label consistency metrics for a fair comparison.
>
> | K | Prec (CS / Ours) | Maj (CS / Ours) | Hit (CS / Ours) |
> |---|---|---|---|
> | 1 | .809 / .872 | .809 / .872 | .809 / .872 |
> | 2 | - | - | .809 / .948 |
> | 3 | - | - | .968 / .970 |
> | 5 | .798 / .854 | .842 / .901 | .968 / .987 |
> | 10| .789 / .844 | .872 / .921 | .990 / .997 |
> | 20| .778 / .832 | .868 / .910 | - |
> | 50| .762 / .813 | .862 / .885 | - |
>
> KREPES outperforms CS by identifying influential instances via representer landmarks, rather than relying on raw geometric proximity which may capture spurious structure.
> # W2: Justification of Kernel-Based Interpretability in SSL
> Our goal is not to compare interpretability of kernels vs. neural networks, but to develop practical, quantitative metrics (e.g., influence functions) for interpreting SSL representations. While such metrics exist in principle for neural networks, they are intractable due to the inverse Hessian.
>
> Existing methods are largely theoretical and limited to linear models. By approximating NNs with eNTK, we make influence functions tractable and enable systematic interpretation of SSL models. This is particularly evident in our auditing experiments, where kernel-based influences allow us to detect and quantify effects such as gender bias in representations
>
> We conducted Inference-Time Latent Ablation to test reliance on KREPES landmarks. Using a 50-NN classifier on frozen embeddings, we removed the top-10 KREPES-identified landmarks and compared the confidence drop against random deletions on the full test set.
>
> | Dataset (N) | Base Conf | Drop (Rand / KREPES) |
> |---|---|---|
> | Adult (200k) | .7542 | .0002 / **.0453** |
> | CIFAR-10 (10k)| .8126 | .0011 / **.0651** |
>
> While the deep model is entirely robust to random data deletion, removing the exact landmarks identified by KREPES causes a confidence collapse: a direct proof that KREPES isolates the true causal support vectors of the deep model. An implication is **Data curation** via pruning samples with near-zero global influence score. Other implications
>
>  - Targeted Augmentation Design: suppress spurious features (e.g., background bias)
>  - Hard Negative Mining: identify and up-weight hard negatives in SimCLR loss
>  - Label-Efficient Fine-Tuning: label most influential samples
> # W3 & Key Q2: Performance–Interpretability Trade-off & Verification via Solo-learn
> While undesirable, 2% accuracy drop reflects a standard and expected trade-off in interpretable models [1,4,5], rather than a critical flaw, as our focus is interpretability rather than SOTA performance.
>
> Following reviewer’s suggestion, we reproduced results using solo-learn CIFAR-10 + Barlow Twins parametrizations, improving downstream accuracy from 90.31% to 91.18%.
> # Minor Weakness & Key Q1: Clarity of Presentation
> Concept Activation Vectors in SSL are detailed in Appendix K; concept sets (e.g., “Sea”) are built from external datasets (e.g., CIFAR-100) to compute CAVs. While SSL training is unsupervised, evaluation and interpretation commonly use supervised signals. Our Conceptual Influence Profile follows this paradigm: models are trained without labels, while human-defined concepts are used to interpret the learned representation space.
>
> The claim of insufficient motivation is very general; we ask the reviewer to specify sections that require clarification so we can improve the presentation.
> # References: Please see response to reviewer 2e3x

---

> > ### Author Rebuttal · Reviewer_PcWw · 2026-04-04
> >
> > I thank the authors for the detailed and thoughtful rebuttal. Both the tables addressed my concern regarding more standardized evaluation setups. I understand and accept the tradeoff between SOTA and interpretable models. The data curation and pruning table is certainly a more direct evaluation regarding feature importance (I would not use the word causal support vectors since it requires insertion/deletion experiments and this seems only deletion). As such I am increasing my score.
> >
> > I agree about the point of instance-level vs dataset-level techniques. However, the definition of interpretability, as defined by the authors is quite narrow since clustering can directly be used to define instance-level distances. I have gone through the other reviews as well and believe that some of the claims in the paper can be tempered.

---

> > > ### Author Response · Authors · 2026-04-07
> > >
> > > We thank the reviewer for increasing the score. We agree with the distinction regarding insertion/deletion experiments and rephrase "causal support vectors" as "highly influential training samples".
> > >
> > > To clarify why KREPES provides interpretability fundamentally distinct from, and more expressive than, instance-level clustering distances, we provide additional empirical evidence in the following (anonymized single-page, figure-only) PDF containing the plots referenced in this response: https://figshare.com/s/5756a99417a9acbf86d4
> > > 1. **Optimization Influence vs. Spatial Proximity (Plot b):**
> > > Clustering metrics (e.g., $k$-NN) assume a test point's nearest geometric neighbors are its most explanatory samples. Plot (b) demonstrates this is insufficient for understanding SSL optimization. For a given test query e.g., the orange star corresponding to the automobile shown in Plot (c), KREPES identifies highly influential landmarks that originate from further apart across the manifold, rather than within its immediate local cluster. KREPES correctly isolates the structural anchors that drove the optimization of the SSL loss (as the influence score originates from the gradient and $H^{-1}$), ignoring geometrically close but optimization-redundant samples.
> > > 2. **Quantifying Negative Influence (Plot c):**
> > > KREPES captures negative influence—identifying specific training landmarks that exerted repelling gradient forces on the test representation during training. Distance-based clustering metrics are strictly non-negative and are incapable of surfacing these oppositional optimization dynamics.
> > > 3. **Label-Free Hyperparameter Tuning via Spectral Diagnostics (Plots e & f):**
> > > Beyond post-hoc attribution, KREPES encodes the structure of the SSL representation space. Plots (e) and (f) track the Spectral Entropy of the  ($\tilde{A}$) across a logarithmic grid of the Barlow Twins regularization parameter ($\lambda$). The peak of this purely unsupervised spectral entropy aligns perfectly with the optimal linear probe accuracy. This demonstrates that KREPES can be utilized as a rigorous label-free diagnostic for SSL model selection, addressing a major practical limitation in unsupervised learning pipelines.
> > >
> > > **Extended Results on CIFAR-10 (solo-learn)**\
> > > We further evaluated KREPES across additional losses using the solo-learn library, following its standard settings (hyperparameters and augmentations). In addition to the 10% linear probing, we also report results with 100% labeled data, as is the standard in SSL:
> > > ### Table 1: Downstream Accuracy of Barlow Twins on CIFAR-10 (% Labels)
> > > | Loss | Ours (10%) | Ours (100%) | Solo-learn (100%) |
> > > |-|-|-|-|
> > > |BT| 90.31| 90.62| 91.18|
> > > |VICReg | 89.88| 90.35| 90.97|
> > > |SimCLR| 89.54| 90.13| 90.36|
> > > |BYOL| 88.76| 89.03| 90.56|
> > >
> > > These confirm that the performance ceiling of our interpretable surrogate is governed by underlying deep SSL backbone, closing the previously observed performance gaps.

---

### Decision · Program_Chairs · 2026-04-30

**Decision:**

Accept (regular)

**Comment:**

This paper introduces KREPES, a framework for interpretable self-supervised learning that uses empirical neural tangent kernel approximations with the Representer Theorem to identify influential training examples ("Representer Landmarks") and quantify representational transparency. A Nystrom approximation enables scalability to datasets with over 1M samples across SSL objectives including SimCLR, BYOL, and VICReg.

On the positive side, reviewers found the problem important and the approach novel. There were no concerns about soundness. The reviewers broadly praised the experimental validation. (One reviewer was initially concerned about the experiments, but this was resolved in rebuttal.)

On the other hand, there are concerns about the positioning and approach that reviewers struggled with. Reviewer PcWw was concerned that the paper relies on unexamined assumptions about what constitutes "interpretable". More than one reviewer was concerned that this method only provides interpretability for the kernel and not the neural network itself.

Although all concerns were not fully addressed during rebuttal, I think this is a solid direction. As the authors point out, similar approaches have been used in supervised learning interpretability, which suggests that this paper will be an important paper for SSL interpretability.